# Theoretical Investigation on Inductive Bias of Isolation Forest

Qin-Cheng Zheng [1 2]   Shao-Qun Zhang [1 3]   Shen-Huan Lyu [4 5]   Yuan Jiang [1 2]   Zhi-Hua Zhou [1 2 †]

## Abstract

Isolation Forest (iForest) is one of the most widely used unsupervised anomaly detectors, owing to its efficiency and performance on large-scale tasks. Despite its broad applications, there is still a lack of theoretical understanding of iForest's empirical success. In this work, we study the inductive bias of iForest and examine when and to what extent it performs well. The main idea is to characterize the random growth process of iForest, in which both split dimensions and split values are selected randomly. We model the growth process of iForest as a random walk and derive the expected path length function, the outcome of iForest that determines the anomaly score, by analyzing the hitting time of the absorbing state. The infinite-sample size analysis reveals that, unlike $k$-Nearest Neighbor ($k$-NN), whose score reflects only the local density, the iForest path length combines the density and the centrality. Since central points naturally have larger path lengths, iForest is therefore less sensitive to central anomalies. Analyses of fixed datasets corroborate this finding and further show that iForest is more parameter-adaptive than $k$-NN. Our study provides a theoretical understanding of the effectiveness of iForest and establishes a foundation for further exploration.

## 1. Introduction

Unsupervised anomaly detection is a fundamental problem in the fields of machine learning and data mining. It is widely used in many real-world tasks, including fraud detection (Fawcett & Provost, 1999; Chandola et al., 2009), network intrusion detection (Phoha, 2002), and medical diagnosis (Wong et al., 2003; Fernando et al., 2022). The goal is to identify anomalies in the concerned data set, which are "few and different" from normal samples (Liu et al., 2008; 2012). Among various types of unsupervised anomaly detection methods, iForest (Liu et al., 2012) stands out as one of the most popular choices. The basic idea is to gather a set of Isolation Trees (iTrees) trained from the observed data, and the anomaly score of a data point is defined as the average path length from the root node to the leaf node of the iTrees. The core concept of iTrees lies in uniformly randomly selecting an attribute and a split value, partitioning the data points into two subsets iteratively, and repeating this process until all data points are isolated. Although highly heuristic, iForest is usually the first choice for anomaly detection. iForest outperforms other unsupervised anomaly detectors in many real-world applications in terms of accuracy, efficiency, and large-scale applicability in practice (Liu et al., 2008; 2012; Aggarwal, 2017; Pang et al., 2019; Cook et al., 2020; Pang et al., 2022).

Despite notable progress in practical applications, the theoretical understanding of the iForest algorithm remains limited. Several seminal works have studied topics such as convergence (Siddiqui et al., 2016), PAC theory (Liu et al., 2018), and learnability (Fang et al., 2022) in the context of anomaly detection. However, these investigations are typically not specific to iForest and thus do not provide a comprehensive explanation of the algorithm. In addition, Scornet et al. (2015); Gao & Zhou (2020); Gao et al. (2022) studied the consistency of random forests, whereas random forests focus on supervised learning tasks and have a different structure from iForest. More recently, Pelletier (2024) showed that iForest path length is $\Theta(\log n)$ in the infinite-sample limit. However, their analysis is too coarse to identify which points have longer or shorter paths, or the components shaping the path-length function. Thus, a theoretical framework tailored to iForest, along with comparisons to other anomaly detectors, remains crucial.

This paper aims to advance the theoretical understanding of iForest by investigating the inductive bias, i.e., the conditions under which it works and the extent of its efficacy. The algorithm's inherently stochastic growth mechanism, in which each split is selected randomly and can significantly

---

[1]National Key Laboratory for Novel Software Technology, Nanjing University, China [2]School of Artificial Intelligence, Nanjing University, China [3]School of Intelligence Science and Technology, Nanjing University, China [4]College of Computer and Information, Hohai University, China [5]Key Laboratory of Water Resources Big Data Technology of Ministry of Water Resources, Hohai University, China. Correspondence to: Zhi-Hua Zhou <zhouzh@lamda.nju.edu.cn>.

*Proceedings of the 43$^{rd}$ International Conference on Machine Learning*, Seoul, South Korea. PMLR 306, 2026. Copyright 2026 by the author(s).

alter the structure of an Isolation Tree, presents considerable challenges in evaluating its outcomes and identifying the inductive bias. To address this issue, we model the growth process of iTrees as a random walk model. This perspective helps derive the closed-form expression of the expected path length function of iForest, the output of iForest that serves as the anomaly detection criterion. The expected path length function enables analysis of the inductive bias and provides a basic framework for understanding iForest. The closed-form expression also allows us to derive an integral-form expression for infinite-sample settings.

Our contributions can be summarized as follows:

1. We open the black box of iForest by modeling its growth process as a random walk and deriving an exact path length function for theoretical analysis.

2. We characterize the infinite-sample inductive bias of the path length function, showing that iForest reflects both density and centrality within the data support.

3. We examine the fixed-dataset inductive bias through case studies against $k$-NN, revealing that iForest is less sensitive to central anomalies and more adaptive.

The rest of this paper is organized as follows. Section 2 introduces notation, settings, and concepts. Section 3 presents the novel random walk view of the growth process of iForest, the derived exact path length function, and a projected multi-dimensional construction. Section 4 analyzes the path length function as the sample size tends to infinity. Section 5 analyzes the fixed-dataset inductive bias of iForest through case studies. Section 6 conducts empirical studies to verify the theoretical findings. Section 7 concludes the work. All proofs are deferred to the Appendix.

## 2. Preliminary

**Settings.** Consider the unsupervised anomaly detection setting. Let $\mathcal{X} \subset \mathbb{R}^d$ and $\mathbf{x} = (\mathbf{x}^{(1)}, \ldots, \mathbf{x}^{(d)}) \in \mathcal{X}$ be the input space and a $d$-dimensional vector, respectively. We observe $D = \{\mathbf{x}_1, \mathbf{x}_2, \ldots, \mathbf{x}_n\}$ consisting of $n$ samples drawn from a distribution $\mathcal{D}$ on $\mathcal{X}$. Dataset $D$ contains $n_0$ normal samples and $n_1$ anomalies where $n = n_0 + n_1$. There is an unknown and unlabeled anomaly subset $A \subsetneq D$, and we are required to identify the anomaly subset $A$.

In unsupervised anomaly detection tasks, however, ground-truth labels are unavailable, making such tasks inherently subjective. Different individuals may consider different anomalies, depending on one's preferences. For such tasks, there are no loss functions to optimize, and understanding the inductive biases of various anomaly detection algorithms is crucial, rather than focusing solely on their absolute performance. In this study, we compare the inductive biases of two widely used anomaly detectors: iForest and $k$-NN.

---

**Algorithm 1** BuildTree($D$)

**Input**: A dataset $D = \{\mathbf{x}_1, \mathbf{x}_2, \ldots\}$
**Output**: An Isolation Tree $T$

1: **if** $|D| \leq 1$ **then**
2:     **return** Leaf
3: **end if**
4: $j \leftarrow$ uniform random in $\{j \mid \#\{\mathbf{x}^{(j)}\} > 1\}$
5: $s \leftarrow$ uniform random in $[\min \mathbf{x}^{(j)}, \max \mathbf{x}^{(j)}]$
6: $D_{\text{left}} \leftarrow \{\mathbf{x} \in D \mid \mathbf{x}^{(j)} \leq s\}$
7: $D_{\text{right}} \leftarrow \{\mathbf{x} \in D \mid \mathbf{x}^{(j)} > s\}$
8: Node $\leftarrow \{$SplitAtt $\leftarrow j$
               SplitValue $\leftarrow s$
               Left $\leftarrow$ BuildTree($D_{\text{left}}$)
               Right $\leftarrow$ BuildTree($D_{\text{right}}$)$\}$
9: **return** Node

---

**Algorithm 2** PathLength $(\mathbf{x}, T)$

**Input**: A sample $\mathbf{x}$ and an Isolation Tree $T$
**Output**: The path length of $\mathbf{x}$ in $T$

1: **if** $T$ is a Leaf **then**
2:     **return** 0
3: **end if**
4: $j \leftarrow T.\text{SplitAtt}, s \leftarrow T.\text{SplitValue}$
5: **if** $\mathbf{x}^{(j)} \leq s$ **then**
6:     **return** $1 + \text{PathLength}(\mathbf{x}, T.\text{Left})$
7: **else**
8:     **return** $1 + \text{PathLength}(\mathbf{x}, T.\text{Right})$
9: **end if**

---

**iForest.** The concept of iForest (Liu et al., 2008) has expanded beyond anomaly detection and become a learning framework for diverse machine learning tasks, including density estimation (Ting et al., 2021), time-series analysis (Ting et al., 2022), kernel learning (Ting et al., 2018; Xu et al., 2019; Ting et al., 2020), and others. There are also many variants of iForest, such as SCiForest (Liu et al., 2010), LSHiForest (Zhang et al., 2017), EIF (Hariri et al., 2021), and Deep Isolation Forest (Xu et al., 2023). Note that theoretically understanding the large family of isolation-based methods remains a long-term challenge. This paper focuses on the original iForest, which is the basis for understanding the idea of isolation for the algorithm family.

iForest works under the belief that anomalies are easier to isolate than normal data points if we uniformly randomly partition the whole feature space until all the points are isolated. iForest constructs a collection of iTrees independently, and the anomaly score is negatively correlated with the average path length from the root node to the leaf node. The procedure of iForest is detailed in Algorithms 1–3.

For convenience, we introduce two simplifications: relaxing the height limit of iTrees and directly outputting the aver-

---

**Algorithm 3** Isolation Forest

---

**Input**: A dataset $D$ and the number of trees $M$
**Output**: A score function
1: **for** $m = 1, \ldots, M$ **do**
2:     $h_m \leftarrow \text{PathLength}(\cdot, \text{BuildTree}(D))$
3: **end for**
4: **return** $M^{-1} \sum_{m=1}^{M} h_m$

---

age path length (Liu et al., 2008) instead of computing the anomaly score. For the former simplification, constraining the height limit will not alter the relative order of isolation path lengths and is usually applied to detect multiple anomaly candidates (Liu et al., 2008), which is not the focus of this work. Regarding the latter simplification, note that the anomaly score introduced by Liu et al. (2008)

$$s(\mathbf{x}, n) = 2^{-\mathbb{E}_{\Theta}[h(\mathbf{x}; D, \Theta)]/c(n)} ,$$

represents a monotonically decreasing transformation of the path length, where $c(n)$ is a normalization constant. This shifts the analysis from anomaly scores to path lengths without altering the fundamental nature of the problem.

$k$-**NN anomaly detectors.** Nearest neighbor-based methods can be roughly separated into two categories: density-based methods that assume the anomalies are in low-density regions and distance-based methods that assume the anomalies are far from normal points. Among the former, the most popular ones include Local Outlier Factor (Breunig et al., 2000). Among the latter, the most typical ones define the anomaly score as the distance to the $k$-th nearest neighbor of a point (Byers & Raftery, 1998; Guttormsson et al., 1999; Sugiyama & Borgwardt, 2013; Ting et al., 2017; Gu et al., 2019). Here, we consider the most standard $k$-NN anomaly detector, which defines the anomaly score as the average distance to the $k$ nearest neighbors. Formally, the scoring function is defined by

$$h_{knn}(\mathbf{x}; D) \triangleq \frac{1}{k} \sum_{\mathbf{x}' \in \mathcal{N}_k(\mathbf{x})} \|\mathbf{x} - \mathbf{x}'\|_1 ,$$

where $\mathcal{N}_k(\mathbf{x})$ is the set of the $k$ nearest neighbors of $\mathbf{x}$ in $D$. The choice of the $L_1$-norm is representative because all norms in finite-dimensional spaces are equivalent. The reason we select the $L_1$-norm is that each iTree partitions the entire feature space into multiple hyper-rectangles. This characteristic aligns more closely with the $L_1$-norm than the more commonly used $L_2$-norm.

Here, we employ the distance-based $k$-NN anomaly detector rather than density-based approaches that score samples by the density function $p(x)$. It becomes apparent that density-based algorithms employing the rectangle kernel $K(u) = 2^{-d} \mathbb{I}(\|u\|_1 \leq 1)$ tend to be equivalent to $k$-NN with the $L_1$-distance. This equivalence arises due to

the strong correlation between the number of points in the neighborhood of a point and the distance to its neighbors.

**Notations.** We denote by $f(n) = O(g(n))$ and $f(n) = \Omega(g(n))$ if there exist constants $c_1, c_2 > 0$ such that $f(n) \leq c_1 g(n)$ and $f(n) \geq c_2 g(n)$, respectively, for all sufficiently large $n \in \mathbb{N}^+$. Similarly, $f(n) = o(g(n))$ and $f(n) = \omega(g(n))$ if $f(n)/g(n) \to 0$ and $f(n)/g(n) \to \infty$ as $n \to \infty$, respectively. Furthermore, $f(n) = \Theta(g(n))$ if both $f(n) = O(g(n))$ and $f(n) = \Omega(g(n))$ hold simultaneously. Let $x_{i:j}$ denote the vector $(x_i, \ldots, x_j)$.

## 3. Path Length Function of iForest

In this section, we analyze the path length function, which is the scoring criterion of iForest and serves as a basis for our subsequent inductive bias analysis. We model the construction of an iTree as a random walk, derive a closed-form expression for the path length function in the one-dimensional case, and extend the analysis to a projected multidimensional iTree construction.

### 3.1. The random walk model for iTrees

We denote the average path length of a point $\mathbf{x}$ in an iTree as $h(\mathbf{x}; D, \Theta)$, where $\Theta$ consists of all the randomness from the construction of the iTree, including the randomness of attribute selection and split-value sampling. To begin with, we introduce the following conclusion.

**Proposition 3.1** (Concentration of iForest). *For any fixed dataset $D$ and any $\mathbf{x} \in \mathcal{X}$, we have*

$$\Pr\left[\left|\frac{1}{M} \sum_{m=1}^{M} h(\mathbf{x}; D, \Theta_m) - \mathbb{E}_{\Theta}[h(\mathbf{x}; D, \Theta)]\right| \geq \epsilon\right]$$
$$\leq 2 \exp\left(-2\epsilon^2 M/n^2\right) ,$$

*where $h(\mathbf{x}; D, \Theta_m)$ is the path length of $\mathbf{x}$ in the $m$-th tree.*

Proposition 3.1 shows that the empirical mean of the path length function $M^{-1} \sum_{m=1}^{M} h(\mathbf{x}; D, \Theta_m)$ converges with high probability to its expectation $\mathbb{E}_{\Theta}[h(\mathbf{x}; D, \Theta)]$ as the number of trees $M$ tends to infinity. Following the proposition, it suffices to analyze the expected path length function $\mathbb{E}_{\Theta}[h(\mathbf{x}; D, \Theta)]$. However, the randomness of the generating process of iTrees poses a challenge, as every split is chosen randomly but depends on the dataset, and a different split can lead to a completely different tree.

Next, we demonstrate the random walk model of the growth process of iTrees. We begin with the case of $d = 1$, i.e., $x_i \in \mathcal{X} = \mathbb{R}$. For simplicity, we assume that $x_1 < x_2 < \cdots < x_n$ is sorted in ascending order. Suppose we care about the path length of $x_i$, each tree node forms an interval, and let $x_{\ell_t}$ and $x_{r_t}$ denote the endpoints of the interval containing $x_i$ at time $t$. This interval-based view removes the irrelevant

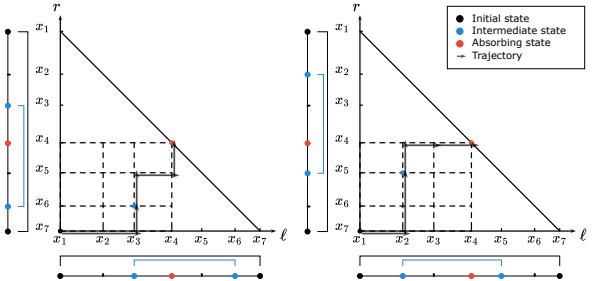

*Figure 1.* Examples of the random walk model for iTrees. The points colored in black, blue, and red indicate the initial, intermediate, and absorbing states, respectively. The lines with arrows indicate the trajectory of the transition process.

parts of the tree and records only the current range that still contains the target point. A split outside this range does not affect the subsequent path of $x_i$, whereas a split inside it updates either the left or the right endpoint. Therefore, the dynamics of the interval can be fully described by the two boundary indices. The following theorem shows that $\mathbf{s}_t \triangleq (x_{\ell_t}, x_{r_t})$ induced by iTrees is a random walk.

**Theorem 3.2** (Random Walk Model for iTree). *For any* $x_\ell, x_r, x_{\ell'}, x_{r'} \in D$, *random process* $\mathbf{s}_t(x_i) = (x_{\ell_t}, x_{r_t})$ *is a random walk with transition probability*

$$\Pr[\mathbf{s}_{t+1} = (x_{\ell'}, x_{r'}) \mid \mathbf{s}_t = (x_\ell, x_r)]$$
$$= \begin{cases} 1, & \text{if } \ell' = \ell = i = r' = r \text{ ,} \\ \dfrac{x_{\ell'} - x_{\ell'-1}}{x_r - x_\ell} \text{ ,} & \text{if } \ell' > \ell, r' = r \text{ ,} \\ \dfrac{x_{r'+1} - x_{r'}}{x_r - x_\ell} \text{ ,} & \text{if } \ell' = \ell, r' < r \text{ ,} \\ 0 \text{ ,} & \text{otherwise .} \end{cases} \quad (1)$$

Figure 1 illustrates the random walk model for iTrees using a seven-point dataset $D = \{x_1, x_2, \ldots, x_7\}$. The left and right subfigures depict two possible trajectories of the random walk. Each state corresponds to a point in a two-dimensional coordinate system, representing the leftmost and rightmost boundaries of the tree node containing the target point. The initial state is $\mathbf{s}_0 = (x_1, x_7)$, shown in black. In each step, the walk transitions rightward or upward, remaining to the left and below the absorbing state $(x_4, x_4)$, which is marked in red. Rightward and upward movements correspond to splitting to the left and right of $x_i$, respectively. The probability of each step is proportional to the distance between consecutive points, as defined in Eq. (1). For instance, as illustrated in the left panel of Figure 1, the random walk traverses from $(x_1, x_7)$ to $(x_3, x_6)$ by moving rightward (with probability $\frac{x_3 - x_2}{x_7 - x_1}$) and subsequently upward (with probability $\frac{x_7 - x_6}{x_7 - x_3}$). The process continues until the absorbing state is reached.

Although we only show the random walk model when $d = 1$,

the model still holds for any $d > 1$. This is because the candidate split attribute and split value only depend on the current point positions, no matter how the state is reached, i.e., the growth of iTrees satisfies the Markov property.

### 3.2. Expected path length of a data point

With the random walk model at hand, the analysis of the expected path length function is now tractable. We begin with a simpler notation as follows

$$\bar{h}(x_i; x_1, \ldots, x_n) \triangleq \mathbb{E}_{\boldsymbol{\Theta}}[h(x_i; x_1, \ldots, x_n, \boldsymbol{\Theta})] \text{ ,}$$

where $x_i \in D$. We first introduce the following lemma, which can simplify the analysis.

**Lemma 3.3.** *For any given dataset $D$ with sample size $n \geq 3$ and $x_i \in D$ for $1 < i < n$, we have*

$$\bar{h}(x_i; x_1, \ldots, x_n)$$
$$= \bar{h}(x_i; x_1, \ldots, x_i) + \bar{h}(x_i; x_i, \ldots, x_n) \text{ .}$$

Lemma 3.3 shows that the expected path length of a data point $x_i$ can be decomposed into two parts, in which the concerned data point is the rightmost and the leftmost point, respectively. From the random walk perspective, the number of steps required to reach the absorbing point $(x_i, x_i)$ equals the sum of the steps taken in the rightward direction and the steps taken upward independently. To be more specific, although opting to move rightward can increase the probability of moving upward in the next step, the probability distribution of the upward step size, conditional on moving upward, remains unchanged; thus, the expected total number of steps remains unchanged, too.

From the above analysis, it suffices to analyze the expected path length of the rightmost point $x_i$ in $\{x_1, \ldots, x_i\}$, as the leftmost point can be analyzed similarly. By the transition rule of random walks, the path length of point $x_i$ is the hitting time of the absorbing state $(x_i, x_i)$. If $T_k$ denotes the expected hitting time starting from state $(x_k, x_i)$, then

$$T_i = 0 \text{ ,}$$
$$T_k = 1 + \sum_{m=k+1}^{i} \frac{x_m - x_{m-1}}{x_i - x_k} T_m \text{ ,} \quad 1 \leq k < i \text{ .}$$

Solving this backward recursion gives the closed-form expression of the expected path length function below.

**Lemma 3.4.** *For any given dataset $D = \{x_1, \ldots, x_i\}$ with sample size $i > 1$, we have*

$$\bar{h}(x_i; x_1, \ldots, x_i) = \sum_{j=2}^{i} \frac{x_j - x_{j-1}}{x_i - x_{j-1}} \text{ ,}$$

$$\bar{h}(x_1; x_1, \ldots, x_i) = \sum_{j=2}^{i} \frac{x_j - x_{j-1}}{x_j - x_1} \text{ .}$$

---

**Algorithm 4** Projected Isolation Forest

---

**Input**: A dataset $D = \{\mathbf{x}_1, \ldots, \mathbf{x}_n\} \subset \mathbb{R}^d$ containing $d$ dimensions points, and the number of trees $M$

**Output**: A projected score function

1: **for** $m = 1, \ldots, M$ **do**
2:     $J_m \leftarrow$ uniform random in $\{1, \ldots, d\}$
3:     $D^{(J_m)} \leftarrow \{x_1^{(J_m)}, \ldots, x_n^{(J_m)}\}$
4:     $T_m \leftarrow \text{BuildTree}(D^{(J_m)})$
5:     $h_m(\mathbf{x}) \leftarrow \text{PathLength}(x^{(J_m)}, T_m)$
6: **end for**
7: **return** $M^{-1} \sum_{m=1}^{M} h_m$

---

Lemma 3.4 presents the closed-form expression of the expected path lengths of the rightmost or leftmost point. Based on a straightforward combination of Lemma 3.3 and Lemma 3.4, we have the expected path length of any $x_i$ in $D$ in the following theorem.

**Theorem 3.5.** *Given $x_i \in D$ with $n > 2$, we have*

$$\bar{h}(x_i; x_1, \ldots, x_n) = \sum_{j=2}^{i} \frac{x_j - x_{j-1}}{x_i - x_{j-1}} + \sum_{j=i+1}^{n} \frac{x_j - x_{j-1}}{x_j - x_i} .$$

Theorem 3.5 presents the closed-form expression for the path length function of any $x_i \in D$. This formula represents the first theoretical characterization of the output of iForest, providing a basis for inductive bias analysis.

For any point outside the dataset, the path length function is the linear interpolation of the path lengths of the two nearest data points. This is a byproduct of this paper and is beyond the scope of inductive bias analysis. For readers interested in this result, we refer to Theorem A.2, where we provide the detailed conclusion and proof.

### 3.3. Multidimensional Cases

The expected path length function in Theorem 3.5 focuses on the one-dimensional case. We extend the fixed-dataset analysis to multidimensional data by introducing projected iForest, a projected variant of iForest. Algorithm 4 gives the full procedure. Each tree samples a coordinate $J$ uniformly from $\{1, \ldots, d\}$ and builds a one-dimensional isolation tree on the projected data $D^{(J)}$. We denote the resulting expected path length by $\bar{h}_{\text{proj}}(\mathbf{x}; D)$.

We have the following theorem.

**Theorem 3.6.** *Let $D = \{\mathbf{x}_1, \ldots, \mathbf{x}_n\} \subset \mathbb{R}^d$, with projections $D^{(i)} = \{x_1^{(i)}, \ldots, x_n^{(i)}\}$. For any $\mathbf{x} \in \mathbb{R}^d$, the projected expected path length satisfies*

$$\bar{h}_{\text{proj}}(\mathbf{x}; D) = \frac{1}{d} \sum_{i=1}^{d} \bar{h}(x^{(i)}; D^{(i)}) .$$

Theorem 3.6 shows that the projected construction admits an exact dimension-wise decomposition: its expected path length is the average of the one-dimensional expected path lengths over all coordinate projections. Since each tree is grown on a single sampled coordinate, projected iForest avoids the coordinate interactions that make ordinary multi-dimensional iTrees difficult to analyze. This gives a direct way to transfer the one-dimensional formula to a multidimensional path length function.

## 4. Inductive Bias: Asymptotic Analysis

In this section, we analyze the asymptotic behavior of the path length function as the sample size tends to infinity.

### 4.1. Single Bounded Support

We first consider the path length function with a single bounded support. Through an asymptotic analysis in the limit $n \to \infty$, we find that the iForest path length reflects both the density and the centrality of a point within the support. Formally, we have the following.

**Theorem 4.1.** *Let $\mathcal{D}$ be a continuous distribution on $\mathcal{X} \subset \mathbb{R}$ with density function $p(x)$ and bounded support $S = [a, b]$. Suppose that $D = \{x_1, \ldots, x_n\}$ consists of $n$ i.i.d. samples from $\mathcal{D}$. Then, as $n \to \infty$, the path length decomposes into*

$$\mathbb{E}[\bar{h}(x; D)] = \underbrace{\mathsf{P}(x)}_{density} + \underbrace{\mathsf{L}(x)}_{position} + \underbrace{\mathsf{C}(x)}_{constant} + o(1).$$

*Here, $\gamma \approx 0.577$ is the Euler constant, and*

$$\mathsf{P}(x) = \begin{cases} \ln p(a), & x \leq a, \\ 2 \ln p(x), & a < x < b, \\ \ln p(b), & x \geq b, \end{cases}$$

$$\mathsf{L}(x) = \begin{cases} \ln(b - a), & x \leq a, \\ \ln[(x - a)(b - x)], & a < x < b, \\ \ln(b - a), & x \geq b. \end{cases}$$

$$\mathsf{C}(x) = \begin{cases} \ln n + 1 + \gamma, & x \leq a, \\ 2 \ln n + 2 + 2\gamma, & a < x < b, \\ \ln n + 1 + \gamma, & x \geq b. \end{cases}$$

The three terms $\mathsf{P}$, $\mathsf{L}$, and $\mathsf{C}$ are the density, position, and constant contributions, respectively. The term $\mathsf{C}$ collects the sample-size and Euler-constant terms; within each region of the support, it is a constant independent of the specific location of $x$. For any interior point $x \in (a, b)$, the position contribution is $\mathsf{L}(x) = \ln[(x-a)(b-x)]$, where the product $(x - a)(b - x)$ measures how central $x$ sits between the endpoints; we refer to it as the *centrality* of $x$ hereafter. Theorem 4.1 thus shows that, for any interior point, the non-constant part of the path length function of iForest is the log

density plus the log centrality as the sample size tends to infinity. We conclude that iForest is an anomaly detector that differs from nearest neighbor-based methods as well as density-based methods. Note that as $n \to \infty$, $k$-NN scores tend to be equivalent to the inverse of the density function $p(x)$; however, iForest scores consider not only the local information from density, but also the global information from the centrality of $x$ within the support. Theorem 4.1 is consistent with the fixed-dataset decision thresholds later summarized in Table 2, which indicates that anomalies in the central region are less likely to be detected by iForest than by $k$-NN, as a larger decision threshold is required. This also coincides with the smaller path lengths of points with low centrality and the larger path lengths of points closer to the center shown in Liu et al. (2012)[Figure 16].

To verify our finding that iForest scores points by combining the density and the centrality, we plot the density and path length curves of a bimodal Gaussian mixture model in Figure 2. As shown in Figure 2, the density of points near $x = -0.3$ is extremely small, similar to the density in the left and right tails. In contrast, since these points are more central, they have larger path lengths than the tail points.

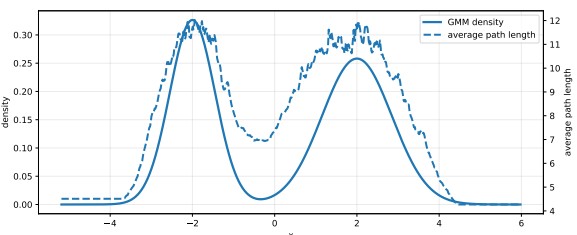

*Figure 2.* Comparison of density and path length.

From Theorem 4.1, the order of $\bar{h}(x; D)$ has terms $\ln n$ for $x \notin (a, b)$, while it has terms $\ln n^2$ for $x \in (a, b)$, as shown in Theorem A.2. This is because for any $x$, even being close to $a$ or $b$, there will be infinitely many samples on the left or right side of $x$. The infinite samples from two sides of $x$ lead to a divergent term $\ln np(x)$.

To clearly show the main component of the path length function, we introduce the following corollary.

**Corollary 4.2.** *Using the same notation as in Theorem 4.1, for any two fixed interior points $x', x'' \in (a, b)$ with $p(x') > 0$ and $p(x'') > 0$, we have*

$$\mathbb{E}[\bar{h}(x'; D) - \bar{h}(x''; D)]$$
$$= 2 \ln \underbrace{\left[ \frac{p(x')}{p(x'')} \right]}_{density\ quotient} + \ln \underbrace{\left[ \frac{(x' - a)(b - x')}{(x'' - a)(b - x'')} \right]}_{centrality\ quotient} + o(1).$$

Corollary 4.2 characterizes the difference between the expected path lengths of two interior points. After subtraction,

the constant term is eliminated, and only the density and centrality contributions remain. Therefore, points with lower density or lower centrality tend to have shorter path lengths, making such points more likely to be identified as anomalies. In contrast, distance- or density-based methods score points according to local density.

### 4.2. Two Bounded Supports

We also have the following corollary.

**Corollary 4.3.** *Using the same notation as in Theorem 4.1, let the support set be $S = [a, b] \cup [c, d]$, where $b < c$. Then,*

$$\mathbb{E}[\bar{h}(x; D)] = \underbrace{\mathsf{P}(x)}_{density} + \underbrace{\mathsf{L}(x)}_{position} + \underbrace{\mathsf{C}(x)}_{constant} + o(1).$$

*The three terms are defined as*

$$\mathsf{P}(x) = \begin{cases} \ln p(a), & x \leq a, \\ 2 \ln p(x), & x \in (a, b) \cup (c, d), \\ \ln p(d), & x \geq d, \end{cases}$$

$$\mathsf{L}(x) = \begin{cases} \ell_a, & x \leq a, \\ \ell_{ab}(x), & x \in (a, b), \\ \ell_{cd}(x), & x \in (c, d), \\ \ell_d, & x \geq d, \end{cases}$$

$$\mathsf{C}(x) = \begin{cases} \ln n + 1 + \gamma, & x \leq a \ or \ x \geq d, \\ 2 \ln n + 2 + 2\gamma, & x \in (a, b) \cup (c, d). \end{cases}$$

*On the gap $[b, c]$, set $\mathsf{C}(x) = \ln n + 1 + \gamma$, while $\mathsf{P}(x)$ and $\mathsf{L}(x)$ linearly interpolate from $\ln p(b)$ to $\ln p(c)$ and from $\ell_b$ to $\ell_c$, respectively. The position entries in $\mathsf{L}$ are*

$$\ell_a = \ln \frac{(b - a)(d - a)}{c - a}, \quad \ell_b = \ln \frac{(b - a)(d - b)}{d - c},$$

$$\ell_c = \ln \frac{(d - c)(c - a)}{c - b}, \quad \ell_d = \ln \frac{(d - c)(d - a)}{d - b},$$

$$\ell_{ab}(x) = \ln[(x - a)(b - x)] + \ln \frac{d - x}{c - x},$$

$$\ell_{cd}(x) = \ln[(x - c)(d - x)] + \ln \frac{x - a}{x - b}.$$

Note that for $x \in (b, c)$, which is outside the support set $[a, b] \cup [c, d]$, the expected path length is the linear interpolation of the expected path lengths of $\mathbb{E}[\bar{h}(b; D)]$ and $\mathbb{E}[\bar{h}(c; D)]$, as shown in Theorem A.2. From Corollary 4.3, we can model the distributions underlying the fixed-dataset marginal anomaly cases studied later in Figures 3 and 5, where the density in $[a, b]$ may be low, and the interval length $(b - a)$ is small, leading to a smaller path length for points in $[a, b]$. Note that for central anomalies, one can model the distribution by a three-part support, i.e., $[a, b] \cup [c, d] \cup [e, f]$, and the expected path length function will be piecewise.

We can similarly derive an asymptotic expression of the expected path length function analogous to Corollary 4.3 with the orders of the fixed-dataset decision thresholds later summarized in Table 2.

# 5. Inductive Bias: Fixed-Dataset Analysis

Based on the closed-form expressions of the expected path length functions, we conduct case studies to investigate iForest's inductive bias by comparing it with $k$-NN. Before that, we introduce the density metrics.

**Definition 5.1.** Let $U \triangleq \max_{i \geq 2} |x_{i+1} - x_i|$ and $L \triangleq \min_{i \geq 2} |x_{i+1} - x_i|$ be the maximum and minimum distances between adjacent points, respectively. We define the density ratio $\kappa \triangleq U/L$ and the density difference $\delta \triangleq U - L$. We say that $D$ is $\kappa$-dense or $\delta$-dense if $D$ has density ratio $\kappa$ or density difference $\delta$, respectively.

The two metrics evaluate how non-uniformly data are distributed relative to their neighbors, measured by ratio and difference. Intuitively, a dataset composed entirely of normal points is expected to exhibit small values for both metrics; otherwise, some points would be excessively distant from their neighbors. These density metrics play a critical role in shaping the inductive bias of iForest and $k$-NN.

For density ratio, we introduce the assumption.

**Assumption 5.2.** Assume that $\kappa \geq \sqrt{n+3}$.

The case studies below are all based on Assumption 5.2. We provide a theoretical study of the mildness of Assumption 5.2 in Appendix C, which can also be verified by computing the density ratio and comparing with the quantity $\sqrt{n+3}$. Table 1 lists the results of checking features of binary classification datasets from the OpenML benchmark, where "Successful" indicates the number and the percentage of features whose density ratio exceeds $\sqrt{n+3}$ while "Valid" denotes those of features with non-repeated values in the dimension. Notice that the probability of repeated values is zero for continuous distributions. Thus, we omit this case for convenience. It is evident that Assumption 5.2 holds since 99.7% of the features meet the condition.

*Table 1.* Number of features of binary classification datasets in the OpenML benchmark that satisfy Assumption 5.2.

| Successful | Valid | Total |
|---|---|---|
| 930,738 (99.7%) | 930,751 (99.7%) | 933,440 (100%) |

## 5.1. Marginal single anomaly

The first case study is about the marginal single anomaly, an example of which is shown in Figure 3, where the anomaly is located at the leftmost position and the normal points are

distributed on the right side. For the rightmost anomaly with normal points on the left, we can obtain a similar conclusion. Marginal single anomalies may be the most common ones in practice, arising when there is an abrupt change in normal data drawn from a continuous and bounded support. Note that although this model looks simple, it is the basic building block of more complex models.

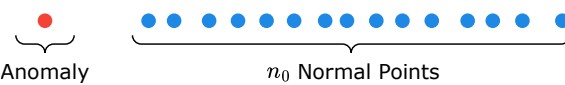

*Figure 3.* Marginal single anomaly.

For the marginal single anomaly, we have the following theorem for iForest.

**Theorem 5.3.** *Suppose that $x_2 - x_1 > U\kappa$. Then, for any dataset $D = \{x_1, \ldots, x_n\}$ such that $x_{2:n}$ is $\kappa$-dense, we have $\bar{h}(x_1; x_{1:n}) < \bar{h}(x_j; x_{1:n})$ for all $j > 1$.*

Theorem 5.3 shows that when $x_2 - x_1 > U \cdot \kappa$, the marginal single anomaly has the smallest expected path length among all data points. Hence, it is the most likely point to be identified as anomalous by iForest. The same conclusion holds for the rightmost marginal single anomaly. Theorem 5.3 shows the sufficiency of $x_2 - x_1 > U \cdot \kappa$. The following theorem shows the necessity.

**Theorem 5.4.** *For any $n > 4$, there exists a choice of $x_{1:n}$ such that $x_{2:n}$ is $\kappa$-dense, $U < x_2 - x_1 \leq U\kappa$, and $\bar{h}(x_1; x_{1:n}) \geq \bar{h}(x_{j_0}; x_{1:n})$ for some $j_0 > 1$.*

Theorem 5.4 shows that if $x_2 - x_1 > U \cdot \kappa$ is violated, even though $x_2 - x_1$ is the largest distance between two consecutive points, it remains possible for $x_1$ not to be the shallowest point, or in other words, iForest may fail to detect the marginal single anomaly.

For $k$-NN, we have the following conclusion.

**Theorem 5.5.** *For $\delta$-dense $x_{2:n}$, $k$-NN detects the marginal single anomaly if and only if $x_2 - x_1 > U + (k-1)\delta/2$.*

Theorem 5.5 reveals that the threshold for $k$-NN depends on the choice of $k$; when $k$ is too large, the marginal single anomaly is likely to be missed, while choosing a small $k$ may ignore some global information. In summary, Theorems 5.3-5.5 imply that isolation-based anomaly detection is more adaptive to marginal single anomalies than $k$-NN.

## 5.2. Central single anomaly

Central single anomaly is another common type of anomaly, as shown in Figure 4. The anomaly is located at the center of the data points, and the normal points are distributed on both the left and right sides. This models the scenario when the distribution consists of multiple clusters, and the anomaly is likely to arise in the gap between different clusters.

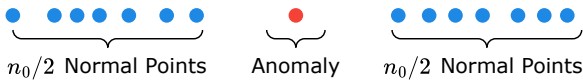

Figure 4. Central single anomaly.

For iForest, we have the following theorem.

**Theorem 5.6.** *Let $n_0$ be even. Suppose both $x_{1:n_0/2}$ and $x_{(n_0/2+2):n}$ are $\kappa$-dense. Then iForest detects $x_{n_0/2+1}$ as the central single anomaly if and only if $\min\{x_{n_0/2+1} - x_{n_0/2}, x_{n_0/2+2} - x_{n_0/2+1}\} > \Theta(\sqrt{n_0\kappa})$.*

Theorem 5.6 shows the sufficiency and necessity for iForest to detect the central single anomaly. Here, the anomaly is "different" enough from the normal points to a degree of order $\Theta(\sqrt{n_0\kappa})$, which is a relatively large quantity, especially when $n_0$ is large. This coincides with Theorem 4.1, which shows that iForest combines density and centrality; thus central anomalies inherently have larger path lengths.

For $k$-NN, we have the following theorem.

**Theorem 5.7.** *Let $n_0$ be even. Suppose both $x_{1:n_0/2}$ and $x_{(n_0/2+2):n}$ are $\delta$-dense. Then $k$-NN detects $x_{n_0/2+1}$ as the central single anomaly if and only if $\min\{x_{n_0/2+1} - x_{n_0/2}, x_{n_0/2+2} - x_{n_0/2+1}\} > \Theta(k\delta)$.*

Theorem 5.7 shows that the threshold for $k$-NN also depends on the choice of $k$, which is algorithm-dependent, while the threshold for iForest relies only on the data distribution, which is problem-dependent. We conclude that iForest is more conservative in detecting central single anomalies but is also more parameter-adaptive than $k$-NN.

### 5.3. Marginal clustered anomalies

Here, we consider the case of marginal clustered anomalies. The motivation is that anomaly events may occur multiple times, as shown in Figure 5. A group of anomalies is located at the leftmost, and the normal points are distributed on the right. Marginal clustered anomalies characterize some unimodal data with multiple clustered noisy points.

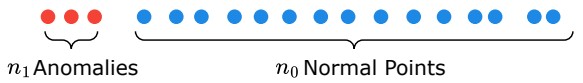

Figure 5. Marginal multiple anomalies.

For iForest, we have the following theorem.

**Theorem 5.8.** *Let $n_1$ be odd with $n_1 = o(n_0)$. For any $D = \{x_1, \ldots, x_{(n_1+n_0)}\}$ such that $x_{1:n_1}$ and $x_{(n_1+1):(n_1+n_0)}$ are $\kappa$-dense, iForest detects all marginal clustered anomalies if and only if $x_{n_1+1} - x_{n_1} > \Theta(n_1^2\kappa)$.*

Theorem 5.8 shows that iForest can detect all the clustered anomalies when normal points are far enough from anomalies, with a considerably large threshold of order $\Theta(n_1^2\kappa)$.

For $k$-NN, we have the following theorem.

**Theorem 5.9.** *Let $k$ be such that $\omega(n_1) \leq k \leq o(n_0)$. Suppose $x_{1:n_1}$ and $x_{(n_1+1):(n_1+n_0)}$ are $\delta$-dense. Then $k$-NN detects all marginal clustered anomalies if and only if $x_{n_1+1} - x_{n_1} > \Theta(k\delta)$.*

Theorem 5.9 shows that the threshold for $k$-NN is also algorithm-dependent, yielding properties similar to those in Theorems 5.5 and 5.7. The choice of $k$ is of great importance for $k$-NN. For instance, when $k$ is too small, $k$-NN may wrongly label the clustered anomalies as normal points, leading to undesirable missed detections and false positives. In contrast, iForest keeps exhibiting adaptability without any hyperparameter tuning. Note that the parameters of iForest are often set to trade off detection performance against computational cost, while those of $k$-NN trade off false positives against false negatives.

Table 2. Decision thresholds of iForest and $k$-NN for different types of anomalies.

| Anomaly Types | iForest | $k$-NN |
|---|---|---|
| Marginal single | $U \cdot \kappa$ | $\Theta(k\delta)$ |
| Central single | $\Theta(\sqrt{n_0\kappa})$ | $\Theta(k\delta)$ |
| Marginal clustered | $\Theta(n_1^2\kappa)$ | $\Theta(k\delta)$ |

Table 2 summarizes three cases of anomalies in this section. It is observed that iForest is more cautious in reporting an anomaly because the threshold is relatively large, but more parameter-adaptive to various types of anomalies than $k$-NN, as the thresholds of iForest are only problem-dependent.

## 6. Empirical Studies

In this section, we conduct experiments to validate our theoretical findings. Specifically, we verify that the empirical path length agrees with our theoretical prediction and examine whether the anomalies detected by iForest are consistent with our theoretical characterization.

### 6.1. Convergence of the empirical path length

This experiment is conducted using both synthetic and real-world datasets. We begin by independently sampling 100 points from three distributions: the standard normal distribution $\mathcal{N}(0, 1)$, the uniform distribution $U[0, 1]$, and the exponential distribution $\text{Exp}(1)$. For real-world datasets, we examine the iris dataset (Fisher, 1936), which is a widely used machine learning dataset. Additionally, we examine the HTTP and SMTP datasets (UCI KDD Archive, 1999), which are widely used anomaly detection datasets. Focusing on the one-dimensional case, we select each feature from all datasets separately to demonstrate the convergence of the empirical path lengths. For each dataset, we evaluate

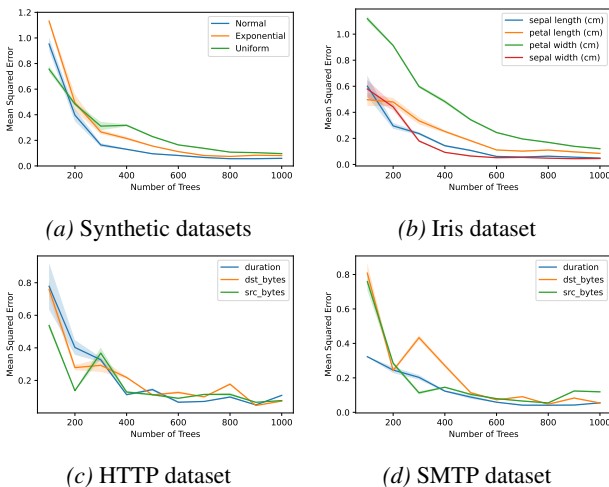

*(a)* Synthetic datasets      *(b)* Iris dataset

*(c)* HTTP dataset      *(d)* SMTP dataset

*Figure 6.* Mean-square errors between the real path lengths learned by iForest and the theoretical expected path lengths as the number of trees increases. The shaded regions represent the confidence regions over multiple runs.

the mean squared errors between the path lengths learned by iForest and the theoretical expected path lengths with 100 subsamples as the number of iTrees ranges from 100 to 1000. Each setup is repeated 10 times independently, and we present the mean and the confidence interval of the mean-squared errors.

Figure 6 shows the error curves with respect to the number of iTrees for different datasets. We observe that for all datasets, the error decreases rapidly and the variance vanishes as the number of trees increases, which implies that the path lengths learned by iForest may eventually converge to the theoretically expected path lengths. This confirms the concentration property of the empirical path lengths around the theoretical path lengths and the correctness of our derived expected path length function, as shown in Proposition 3.1 and Theorem 3.5, respectively.

### 6.2. Anomaly detection of iForest

We generate multiple datasets, each of which contains 20 points drawn i.i.d. from a uniform distribution. We compute the expected path length of each point to help understand when iForest can and cannot detect anomalies. The results are presented in Figure 7, where points with small path lengths are shown in red as anomaly candidates, and points with larger path lengths are shown in blue as normal points.

From Figure 7, three key observations can be made: (1) Marginal single and clustered anomalies are correctly labeled, even though some are not significantly distant from normal points. This aligns with Theorems 5.3 and 5.8, stating that marginal anomalies are more easily isolated. (2) In Figure 7 (a), only the central anomaly is detected.

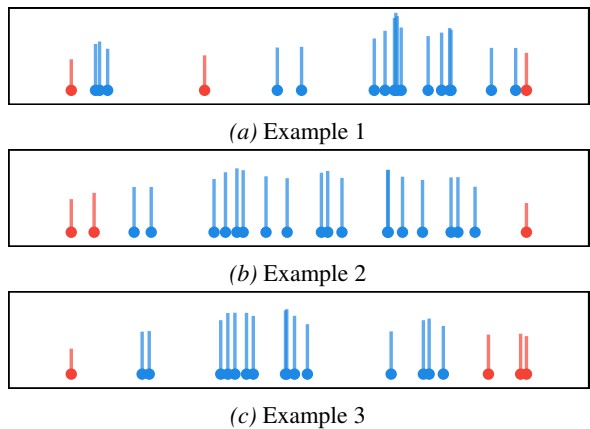

*(a)* Example 1

*(b)* Example 2

*(c)* Example 3

*Figure 7.* Visualization of the expected path lengths for points generated from a uniform distribution. Circles and bars represent the point positions and the expected path lengths, respectively.

This is consistent with Theorem 5.6, which indicates that iForest can identify central anomalies only when they are sufficiently distant from normal data. (3) In Figures 7 (b) and (c), $k$-NN may fail to detect the rightmost clustered anomaly when $k$ is set too small, whereas iForest successfully identifies all marginal clustered anomalies. This supports Theorems 5.8–5.9, demonstrating that iForest is more adaptive to parameter settings than $k$-NN.

## 7. Conclusions

We examined the inductive bias of iForest by modeling its growth process as a random walk and deriving the closed-form expected path length function. Our asymptotic analysis shows that, unlike $k$-NN whose score reflects only the local density, the iForest path length combines the density and the centrality, which intrinsically makes iForest less sensitive to central anomalies. Case studies on fixed datasets corroborate this finding and further show that iForest offers greater parameter adaptivity than $k$-NN.

## Acknowledgments

This research was supported by NSFC (62576165), MoE FIDBP (JYB2025XDXM118), and Jiangsu Science Foundation Leading-edge Technology Program (BK20232003). The authors thank Wei Gao, Jia-Wei Shan, Lue Tao, and Jin-Hui Wu for their helpful discussions.

## Impact Statement

This paper presents work whose goal is to advance the field of Machine Learning. There are many potential societal consequences of our work, none of which we feel must be specifically highlighted here.

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

# A. Proofs in Section 3

Here, we provide the proofs of results in Section 3.

## A.1. Proof of Proposition 3.1

We begin with the following technical lemma.

**Lemma A.1** (Hoeffding's inequality (Hoeffding, 1963)). *Let $X_1, \ldots, X_n$ be independent random variables such that $a_i \leq X_i \leq b_i$ for all $i \in [n]$. Consider the average of these random variables $S_n = \frac{1}{n} \sum_{i=1}^n X_i$. Then, for all $\epsilon > 0$, we have*

$$\Pr\left[|S - \mathbb{E}[S]| \geq \epsilon\right] \leq 2 \exp\left(-2\epsilon^2 / \sum_{i=1}^n (b_i - a_i)^2\right) .$$

Then we prove Proposition 3.1.

**Proof** We first denote $h'(\mathbf{x}; \boldsymbol{\Theta}_m) = M^{-1} h(\mathbf{x}; \boldsymbol{\Theta}_m)$. From the growth process of Isolation Trees, $\boldsymbol{\Theta}_1, \ldots, \boldsymbol{\Theta}_m$ are i.i.d. copies of $\boldsymbol{\Theta}$, and we have

$$\mathbb{E}_{\boldsymbol{\Theta}}[h'(\mathbf{x}; \boldsymbol{\Theta}_m)] = \mathbb{E}_{\boldsymbol{\Theta}}[h'(\mathbf{x}; \boldsymbol{\Theta})], \ \forall 1 \leq m \leq M .$$

By applying Hoeffding's inequality (Lemma A.1) to $h'(\mathbf{x}; \boldsymbol{\Theta}_m)$, we have

$$\begin{aligned}
\Pr\left[\left|\frac{1}{M} \sum_{m=1}^M h(\mathbf{x}; D, \boldsymbol{\Theta}_m) - \mathbb{E}_{\boldsymbol{\Theta}}[h(\mathbf{x}; D, \boldsymbol{\Theta})]\right| \geq \epsilon\right] &= \Pr\left[\left|\sum_{m=1}^M h'(\mathbf{x}; \boldsymbol{\Theta}_m) - \mathbb{E}\left[\sum_{m=1}^M h'(\mathbf{x}; \boldsymbol{\Theta}_m)\right]\right| \geq \epsilon\right] \\
&\leq 2 \exp\left(-2\epsilon^2 / (M(n/M)^2)\right) \\
&= 2 \exp\left(-2\epsilon^2 M / n^2\right) .
\end{aligned}$$

This completes the proof. $\square$

## A.2. Proof of Theorem 3.2

**Markov property.** The Markov property holds directly from the growth process of Isolation Trees.

**Transition probability.** We now verify the transition probability. Recall that the current state, the next state, and the absorbed state are denoted by $(x_\ell, x_r)$, $(x_{\ell'}, x_{r'})$, and $(x_i, x_i)$, respectively, where $x_\ell, x_{\ell'}, x_r, x_{r'}, x_i \in D$, $x_\ell \leq x_i \leq x_r$, and $x_{\ell'} \leq x_i \leq x_{r'}$. In every step, an Isolation Tree chooses the split value in the interval $(x_\ell, x_i)$ or $(x_i, x_r)$, and thus either $x_{\ell'} = x_\ell$ or $x_{r'} = x_r$ holds, implying that

$$\Pr\left[\mathbf{s}_{t+1} = (x_{\ell'}, x_{r'}) \mid \mathbf{s}_t = (x_\ell, x_r)\right] = 0 ,$$

where $x_{\ell'}, x_{r'} \in \{(x, y) \mid (x \neq x_\ell \wedge y \neq x_r) \vee (x = x_\ell \wedge y \geq x_r) \vee (y = x_r \wedge x \leq x_\ell)\}$.

We now consider the case of $x_{\ell'} > x_\ell$ and $x_{r'} = x_r$. The state change from $(x_\ell, x_r)$ to $(x_{\ell'}, x_r)$ happens when the split point is chosen in the interval $(x_{\ell'-1}, x_{\ell'})$. Hence, the transition probability equals the probability of choosing the split point in the interval $(x_{\ell'-1}, x_{\ell'})$, which equals

$$\Pr\left[\mathbf{s}_{t+1} = (x_{\ell'}, x_r) \mid \mathbf{s}_t = (x_\ell, x_r)\right] = \frac{x_{\ell'} - x_{\ell'-1}}{x_r - x_\ell}, \ \forall(x_\ell < x_{\ell'} \leq x_i) .$$

Similarly, for $x_{\ell'} = x_\ell$ and $x_{r'} < x_r$, we have

$$\Pr\left[\mathbf{s}_{t+1} = (x_\ell, x_{r'}) \mid \mathbf{s}_t = (x_\ell, x_r)\right] = \frac{x_{r'+1} - x_{r'}}{x_r - x_\ell}, \ \forall(x_i \leq x_{r'} < x_r) .$$

The proof is complete. $\square$

### A.3. Proof of Lemma 3.3

**Proof**  Let $\Delta_j = x_j - x_{j-1}$. For $\ell \leq i \leq r$, denote by $T_{\ell,r}$ the expected hitting time of the absorbing state $(x_i, x_i)$ when the random walk starts from $(x_\ell, x_r)$. Define the one-sided hitting times

$$A_\ell \triangleq T_{\ell,i} \quad \text{and} \quad B_r \triangleq T_{i,r} .$$

By the random walk transition rule in Theorem 3.2, they satisfy

$$A_i = B_i = 0 , \quad A_\ell = 1 + \sum_{m=\ell+1}^{i} \frac{\Delta_m}{x_i - x_\ell} A_m , \quad \ell < i ,$$

and

$$B_r = 1 + \sum_{m=i}^{r-1} \frac{\Delta_{m+1}}{x_r - x_i} B_m , \quad r > i .$$

For the two-sided random walk, the same first-step analysis gives

$$T_{\ell,r} = 1 + \sum_{m=\ell+1}^{i} \frac{\Delta_m}{x_r - x_\ell} T_{m,r} + \sum_{m=i}^{r-1} \frac{\Delta_{m+1}}{x_r - x_\ell} T_{\ell,m} ,$$

for every $(\ell, r) \neq (i, i)$, with $T_{i,i} = 0$. We now verify that $A_\ell + B_r$ satisfies this two-sided first-step equation:

$$
\begin{aligned}
& 1 + \sum_{m=\ell+1}^{i} \frac{\Delta_m}{x_r - x_\ell}(A_m + B_r) + \sum_{m=i}^{r-1} \frac{\Delta_{m+1}}{x_r - x_\ell}(A_\ell + B_m) \\
&= 1 + \frac{(x_i - x_\ell)(A_\ell - 1) + (x_i - x_\ell)B_r + (x_r - x_i)A_\ell + (x_r - x_i)(B_r - 1)}{x_r - x_\ell} \\
&= 1 + \frac{(x_i - x_\ell)A_\ell - (x_i - x_\ell) + (x_i - x_\ell)B_r + (x_r - x_i)A_\ell + (x_r - x_i)B_r - (x_r - x_i)}{x_r - x_\ell} \\
&= 1 + \frac{(x_r - x_\ell)A_\ell + (x_r - x_\ell)B_r - (x_r - x_\ell)}{x_r - x_\ell} \\
&= 1 + A_\ell + B_r - 1 = A_\ell + B_r .
\end{aligned}
$$

By the uniqueness of the solution to the first-step equations of the finite absorbing random walk, $T_{\ell,r} = A_\ell + B_r$. Taking $(\ell, r) = (1, n)$, we have

$$\bar{h}(x_i; x_1, \ldots, x_n) = T_{1,n} = A_1 + B_n = \bar{h}(x_i; x_1, \ldots, x_i) + \bar{h}(x_i; x_i, \ldots, x_n) ,$$

which completes the proof. $\qquad\square$

### A.4. Proof of Lemma 3.4

**Proof**  We prove the first identity; the second follows from the same argument with the left and right directions reversed. Let $\Delta_j = x_j - x_{j-1}$ for $2 \leq j \leq i$. For $1 \leq k \leq i$, define $T_k$ as the expected hitting time of the absorbing state $(x_i, x_i)$ when the random walk starts from $(x_k, x_i)$. Then $T_i = 0$, and Theorem 3.2 gives the first-step equation

$$T_k = 1 + \sum_{m=k+1}^{i} \frac{\Delta_m}{x_i - x_k} T_m , \quad 1 \leq k < i . \tag{2}$$

We claim that the solution is

$$T_k = \sum_{j=k+1}^{i} \frac{\Delta_j}{x_i - x_{j-1}} , \quad 1 \leq k \leq i ,$$

where the sum is empty when $k = i$. It is clear that $T_i = 0$. For $k < i$, substituting this expression into the right-hand side of Eq. (2) gives

$$1 + \sum_{m=k+1}^{i} \frac{\Delta_m}{x_i - x_k} \sum_{j=m+1}^{i} \frac{\Delta_j}{x_i - x_{j-1}}$$

$$= 1 + \frac{1}{x_i - x_k} \sum_{j=k+2}^{i} \frac{(x_{j-1} - x_k)\Delta_j}{x_i - x_{j-1}}$$

$$= \frac{\Delta_{k+1}}{x_i - x_k} + \sum_{j=k+2}^{i} \frac{\Delta_j}{x_i - x_{j-1}} = \sum_{j=k+1}^{i} \frac{\Delta_j}{x_i - x_{j-1}} .$$

Since this finite random walk is absorbing, the first-step system has a unique solution. Thus,

$$\bar{h}(x_i; x_1, \ldots, x_i) = T_1 = \sum_{j=2}^{i} \frac{x_j - x_{j-1}}{x_i - x_{j-1}} .$$

Applying the same hitting-time argument to the random walk whose right endpoint moves leftward yields

$$\bar{h}(x_1; x_1, \ldots, x_i) = \sum_{j=2}^{i} \frac{x_j - x_{j-1}}{x_j - x_1} .$$

The proof is complete. □

## A.5. Path Length Function of Points Not in the Dataset

For data points not in the dataset, we have the following theorem.

**Theorem A.2.** *For any given dataset $D$ with sample size $n > 2$ and $x \notin D$, we have*

$$\bar{h}(x) = \begin{cases} \bar{h}(x_1) , & \text{if } x < x_1 , \\ \bar{h}(x_n) , & \text{if } x \geq x_n , \\ \ell_i(x) , & \text{if } x_i \leq x < x_{i+1} , \end{cases}$$

*where*

$$\ell_i(x) = \bar{h}(x_i) + \frac{x - x_i}{x_{i+1} - x_i} \left( \bar{h}(x_{i+1}) - \bar{h}(x_i) \right)$$

*is the linear interpolation of $\bar{h}(x_i)$ and $\bar{h}(x_{i+1})$.*

**Proof** For $x < x_1$ or $x \geq x_n$, isolating $x$ is equivalent to isolating $x_1$ or $x_n$, respectively. Thus, we have

$$\mathbb{E}_{\Theta} \left[ h(x; x_1, \ldots, x_n, \Theta) \right] = \mathbb{E}_{\Theta} \left[ h(x_1; x_1, \ldots, x_n, \Theta) \right], \text{for } \forall x < x_1$$

and

$$\mathbb{E}_{\Theta} \left[ h(x; x_1, \ldots, x_n, \Theta) \right] = \mathbb{E}_{\Theta} \left[ h(x_n; x_1, \ldots, x_n, \Theta) \right], \text{for } \forall x \geq x_n .$$

We now consider the case where $x \in [x_1, x_n)$. By Lemma 3.4, it suffices to show that for $x \in [x_i, x_{i+1})$, the expected path length $\mathbb{E}_{\Theta} \left[ h(x; x_1, \ldots, x_n, \Theta) \right]$ is a linear function. For any given $\Theta = \Theta_0$, we have

$$h(x; x_1, \ldots, x_n, \Theta_0)$$

is piecewise-constant. Note that every split that does not isolate $x$ may not change the value of $h(x; x_1, \ldots, x_n, \Theta_0)$. Therefore, we will focus on the split that isolates $x$. Let $\Omega$ denote the event that $x$ is eventually isolated by a split in $(x_i, x)$. Then, we have

$$\mathbb{E}_{\Theta} \left[ h(x; x_1, \ldots, x_n, \Theta) \right] = \Pr[\Theta \in \Omega] \, \mathbb{E}_{\Theta} \left[ h(x; x_1, \ldots, x_n, \Theta) \mid \Theta \in \Omega \right]$$
$$+ \Pr[\Theta \notin \Omega] \, \mathbb{E}_{\Theta} \left[ h(x; x_1, \ldots, x_n, \Theta) \mid \Theta \notin \Omega \right] .$$

When the split isolating $x$ is in interval $(x_i, x)$, point $x$ is treated the same as $x_{i+1}$ and thus has the same path length as $x_{i+1}$. Similarly, when the split isolating $x$ is in interval $(x, x_{i+1})$, point $x$ is treated the same as $x_i$ and thus has the same path length as $x_i$. Therefore, the path length of $x$ is a constant given either $\Theta \in \Omega$ or $\Theta \notin \Omega$, which implies that

$$\mathbb{E}_{\Theta}\left[h\left(x; x_1, \ldots, x_n, \Theta\right)\right] = \Pr[\Theta \in \Omega]c_1 + \Pr[\Theta \notin \Omega]c_2 ,$$

where $c_1$ and $c_2$ are constants. Then it suffices to analyze the probability of the event $\Omega$. By the growth mechanism of Isolation Forest, the split that isolates $x$ is uniformly sampled from the interval $(x_i, x_{i+1})$, and thus we have

$$\Pr[\Theta \in \Omega] = \frac{x - x_i}{x_{i+1} - x_i} \quad \text{and} \quad \Pr[\Theta \notin \Omega] = \frac{x_{i+1} - x}{x_{i+1} - x_i} .$$

As the dataset is fixed, $\Pr[\Theta \in \Omega]$ and $\Pr[\Theta \notin \Omega]$ are both linear in $x$, which completes the proof.

$\square$

### A.6. Proof of Theorem 3.6

Let $J$ be the coordinate sampled by the projected isolation tree. Conditioning on $J = i$, the projected tree is exactly the one-dimensional isolation tree built on $D^{(i)}$, and the path length of $\mathbf{x}$ equals the path length of $x^{(i)}$ in that one-dimensional projected dataset. Therefore, by the law of total expectation,

$$\bar{h}_{\mathrm{proj}}(\mathbf{x}; D) = \frac{1}{d} \sum_{i=1}^{d} \bar{h}(x^{(i)}; D^{(i)}) .$$

The proof is complete.

$\square$

# B. Proof of Section 4

Here, we provide the proofs of results in Section 4.

We begin with the following technical lemma.

**Lemma B.1.** *Let $\mathcal{D}$ be a continuous distribution on $\mathcal{X} \subset \mathbb{R}$ with density function $p(x)$ and bounded support $[a, b]$. Suppose that $D = \{x_1, \ldots, x_n\}$ consists of $n$ i.i.d. samples from $\mathcal{D}$. Without loss of generality, we let $x_1 \leq x_2 \leq \cdots \leq x_n$. Then, the distances between any two consecutive samples vanish as $n \to \infty$, i.e.,*

$$\lim_{n \to \infty} \max_{1 \leq i \leq n-1} |x_{i+1} - x_i| = 0 \quad \text{almost surely} .$$

**Proof** Let $\Delta_n = \max_{1 \leq i \leq n-1}(x_{i+1} - x_i)$ denote the maximum spacing between consecutive order statistics. To prove the lemma, it suffices to show that for any $\epsilon > 0$, $\Delta_n < \epsilon$ almost surely as $n \to \infty$.

Since the support $[a, b]$ is bounded, we can cover it with a finite number of intervals. Fix $\epsilon > 0$. Let $M$ be an integer sufficiently large such that the interval width $\delta = \frac{b-a}{M}$ satisfies $\delta < \frac{\epsilon}{2}$. We partition $[a, b]$ into $M$ disjoint intervals $I_1, I_2, \ldots, I_M$, each of length $\delta$.

Let $q_k = \int_{I_k} p(x)\, dx$ be the probability mass of the $k$-th interval. Since the support is connected and $p(x)$ is a valid density on $[a, b]$, we have $q_k > 0$ for all $k \in \{1, \ldots, M\}$.

Consider the condition under which the maximum gap $\Delta_n > \epsilon$. If there exists a gap between adjacent samples of size greater than $\epsilon$, this gap must necessarily fully contain at least one of the partition intervals $I_k$ (since the length of $I_k$ is $\delta < \epsilon/2$). Therefore, the event $\{\Delta_n > \epsilon\}$ implies that at least one interval $I_k$ contains no samples from $D$.

We can bound the probability of this event using the union bound:

$$P(\Delta_n > \epsilon) \leq P\left(\bigcup_{k=1}^{M} \{\text{no sample in } I_k\}\right) \leq \sum_{k=1}^{M} P(\text{no sample in } I_k).$$

The probability that a specific interval $I_k$ contains no samples out of $n$ independent draws is $(1 - q_k)^n$. Thus:

$$P(\Delta_n > \epsilon) \leq \sum_{k=1}^{M} (1 - q_k)^n.$$

Since $0 < q_k < 1$, the term $(1 - q_k)^n$ decays exponentially with $n$. Consequently, the series sum over $n$ is finite:

$$\sum_{n=1}^{\infty} P(\Delta_n > \epsilon) \leq \sum_{n=1}^{\infty} \sum_{k=1}^{M} (1 - q_k)^n = \sum_{k=1}^{M} \frac{1 - q_k}{q_k} < \infty.$$

By the Borel-Cantelli lemma, the probability that the event $\{\Delta_n > \epsilon\}$ occurs for infinitely many $n$ is 0. Since $\epsilon$ is arbitrary, we conclude that $\lim_{n \to \infty} \Delta_n = 0$ almost surely.

The proof is complete. $\qquad\square$

Lemma B.1 shows that the distances between any two consecutive samples vanish as $n \to \infty$. This is a key condition for the definition of the Riemann integral. From Lemma B.1, we also have the following.

**Lemma B.2.** *Under the same conditions of Lemma B.1, the difference between the path lengths of any two consecutive samples vanishes as $n \to \infty$, i.e.,*

$$\lim_{n \to \infty} \max_{1 \leq i \leq n-1} |\bar{h}(x_{i+1}; D) - \bar{h}(x_i; D)| = 0 \quad \text{almost surely} .$$

**Proof** By the closed form of $\bar{h}(x; D)$ in Theorem 3.5, we have

$$
\begin{aligned}
\bar{h}(x_{i+1}; D) - \bar{h}(x_i; D) = \sum_{j=2}^{i} & \left( \frac{x_j - x_{j-1}}{x_{i+1} - x_{j-1}} - \frac{x_j - x_{j-1}}{x_i - x_{j-1}} \right) \\
+ & \left( \frac{x_{i+1} - x_i}{x_{i+1} - x_i} - \frac{x_{i+1} - x_i}{x_{i+1} - x_i} \right) \\
+ & \sum_{j=i+2}^{n} \left( \frac{x_j - x_{j-1}}{x_j - x_{i+1}} - \frac{x_j - x_{j-1}}{x_j - x_i} \right).
\end{aligned}
$$

Let $\Delta_i := x_{i+1} - x_i$ and $\Delta_n := \max_{1 \leq i \leq n-1} \Delta_i$. By Lemma B.1, we have $\Delta_n \to 0$ almost surely.

For the first summation, for $2 \leq j \leq i$,

$$\left| \frac{x_j - x_{j-1}}{x_{i+1} - x_{j-1}} - \frac{x_j - x_{j-1}}{x_i - x_{j-1}} \right| = (x_j - x_{j-1}) \frac{\Delta_i}{(x_{i+1} - x_{j-1})(x_i - x_{j-1})} \leq \frac{(x_j - x_{j-1})\Delta_i}{(x_i - x_{j-1})^2}.$$

Hence,

$$\sum_{j=2}^{i} \left| \frac{x_j - x_{j-1}}{x_{i+1} - x_{j-1}} - \frac{x_j - x_{j-1}}{x_i - x_{j-1}} \right| \leq \Delta_i \sum_{j=2}^{i} \frac{x_j - x_{j-1}}{(x_i - x_{j-1})^2}.$$

Similarly, for the last summation, for $i + 2 \leq j \leq n$,

$$\left| \frac{x_j - x_{j-1}}{x_j - x_{i+1}} - \frac{x_j - x_{j-1}}{x_j - x_i} \right| \leq \frac{(x_j - x_{j-1})\Delta_i}{(x_j - x_i)^2},$$

and thus

$$\sum_{j=i+2}^{n} \left| \frac{x_j - x_{j-1}}{x_j - x_{i+1}} - \frac{x_j - x_{j-1}}{x_j - x_i} \right| \leq \Delta_i \sum_{j=i+2}^{n} \frac{x_j - x_{j-1}}{(x_j - x_i)^2}.$$

Both sums on the right-hand side are Riemann sums corresponding to integrals of the form

$$\int \frac{1}{(x_i - t)^2} \, dt,$$

which are finite and uniformly bounded. Therefore, there exists a constant $C > 0$ such that

$$\left| \bar{h}(x_{i+1}; D) - \bar{h}(x_i; D) \right| \leq C \Delta_i.$$

Taking the maximum over $1 \leq i \leq n - 1$, we obtain

$$\max_{1 \leq i \leq n-1} \left| \bar{h}(x_{i+1}; D) - \bar{h}(x_i; D) \right| \leq C \Delta_n.$$

Since $\Delta_n \to 0$ almost surely, the conclusion follows. $\square$

### B.1. Proof of Theorem 4.1

1. For any $x \leq a$, by Theorem 3.5 and Theorem A.2, we have

$$\bar{h}(x; D) = \bar{h}(a; D) = \sum_{j=2}^{n} \frac{x_j - x_{j-1}}{x_j - x_1} = 1 + \sum_{j=3}^{n} \frac{\Delta_{j-1}}{\Delta_1 + \cdots + \Delta_{j-1}} \ . \tag{3}$$

By the definition of Riemann integral, we have

$$\sum_{j=3}^{n} \frac{\Delta_{j-1}}{\Delta_1 + \cdots + \Delta_{j-1}} = \int_{x_2}^{x_n} \frac{1}{t - x_1} \, dt + o(1)$$

$$= \ln(x_n - x_1) - \ln(x_2 - x_1) + o(1) \ . \tag{4}$$

By the properties of order statistics (David & Nagaraja, 2004), we have $n(x_2 - x_1) \sim \text{Exp}(p(a))$ and $\ln[np(a)(x_2 - x_1)]$ is a standard Gumbel distribution. Therefore,

$$\mathbb{E}\big[ \ln[np(a)(x_2 - x_1)] \big] \to -\gamma \ , \tag{5}$$

where $\gamma \approx 0.577$ is the Euler constant. By the fact that

$$x_n \xrightarrow{p} b \quad \text{and} \quad x_1 \xrightarrow{p} a \ .$$

Note that $\ln$ is a continuous mapping and that $b - a > 0$, we have

$$\ln(x_n - x_1) \xrightarrow{p} \ln(b - a) \ .$$

Since the $\ln(x_n - x_1)$ is bounded from above by $\ln(b - a)$ and the probability of the range being near zero (where the log diverges) decays rapidly as $n \to \infty$, the sequence is uniformly integrable. Thus, we have:

$$\mathbb{E}[\ln(x_n - x_1)] \to \mathbb{E}\left[\text{plim}_{n \to \infty} \ln(x_n - x_1)\right] = \ln(b - a) \ . \tag{6}$$

Combining Eq. (3)-(6), we have

$$\mathbb{E}[\bar{h}(x; D)] = o(1) + \ln[np(a)(b - a)] + 1 + \gamma \ .$$

2. For any $x \geq b$, by a similar derivation, we have

$$\mathbb{E}[\bar{h}(x; D)] = o(1) + \ln[np(b)(b - a)] + 1 + \gamma \ .$$

3. For any $a < x < b$, note that with probability one, we have $x \notin D$. Therefore, we should use the outer expected path length function in Theorem A.2 to compute the expected path length by linear interpolation, i.e.,

$$\bar{h}(x; D) = \ell_i(x) = \bar{h}(x_i) + \frac{x - x_i}{x_{i+1} - x_i} \left( \bar{h}(x_{i+1}) - \bar{h}(x_i) \right) \ ,$$

where $x_i$ and $x_{i+1}$ are the nearest data points to the left and right of $x$, respectively. By Lemma B.2, we have

$$\mathbb{E}[\bar{h}(x; D)] = \mathbb{E}[\ell_i(x)] = \mathbb{E}[\bar{h}(x_i)] + o(1) . \tag{7}$$

By Theorem 3.5, we have the closed-form expression of $\mathbb{E}[\bar{h}(x_i)]$ as follows:

$$\bar{h}(x_i) = \sum_{j=2}^{i} \frac{x_j - x_{j-1}}{x_i - x_{j-1}} + \sum_{j=i+1}^{n} \frac{x_j - x_{j-1}}{x_j - x_i} .$$

Similar to the analysis of $x \le a$, we can rewrite the above expression by the definition of Riemann integral as follows:

$$\bar{h}(x_i) = \int_{x_{i-1}}^{x_1} \frac{1}{x_i - t} dt + 1 + 1 + \int_{x_{i+1}}^{x_n} \frac{1}{t - x_i} dt + o(1)$$
$$= \ln(x_i - x_1) - \ln(x_i - x_{i-1}) + \ln(x_n - x_i) - \ln(x_{i+1} - x_i) + 2 + o(1) .$$

Similarly, we have

$$\mathbb{E}[\ln(x_i - x_1)] \to \ln(x_i - a)$$
$$\mathbb{E}\big[\ln[np(x_i)(x_i - x_{i-1})]\big] \to -\gamma$$
$$\mathbb{E}\big[\ln[np(x_i)(x_{i+1} - x_i)]\big] \to -\gamma$$
$$\mathbb{E}[\ln(x_n - x_i)] \to \ln(b - x_i) ,$$

which implies that

$$\mathbb{E}[\bar{h}(x_i)] = o(1) + \ln[n^2 p^2(x_i)(x_i - x_{i-1})(x_{i+1} - x_i)] + 2 + 2\gamma .$$

Substituting this into Eq. (7), by the continuity of $\ln$ and $p(x)$, we have

$$\mathbb{E}[\bar{h}(x; D)] = o(1) + \ln[n^2 p^2(x)(x - a)(b - x)] + 2 + 2\gamma .$$

The proof is complete. □

## B.2. Proof of Corollary 4.2

For any fixed interior point $x \in (a, b)$, Theorem 4.1 gives

$$\mathbb{E}[\bar{h}(x; D)] = 2 \ln n + 2 + 2\gamma + 2 \ln p(x) + \ln[(x - a)(b - x)] + o(1).$$

Applying the above to $x'$ and $x''$, and taking the difference, the constant contribution C, including $2 \ln n$ and $2 + 2\gamma$, cancels because it is independent of the interior points. Therefore,

$$\mathbb{E}[\bar{h}(x'; D) - \bar{h}(x''; D)] = 2 \ln p(x') - 2 \ln p(x'') + \ln[(x' - a)(b - x')] - \ln[(x'' - a)(b - x'')] + o(1)$$
$$= 2 \ln \frac{p(x')}{p(x'')} + \ln \frac{(x' - a)(b - x')}{(x'' - a)(b - x'')} + o(1).$$

The proof is complete. □

## B.3. Proof of Corollary 4.3

The proof of Corollary 4.3 follows the proof of Theorem 4.1. The difference is that we have seven cases of the interval that $x$ belongs to: $(-\infty, a], (a, b), \{b\}, (b, c), \{c\}, (c, d), [d, \infty)$.

1. For $x \in (-\infty, a]$, we have

$$\mathbb{E}[\bar{h}(x_1; D)] = \mathbb{E}[\bar{h}(a; D)] = \mathbb{E}[\bar{h}(x_1; D)] + o(1)$$

Let $n' = \max\{i : x_i \le b\}$. Then, we have

$$\bar{h}(x; D) = \sum_{j=2}^{n} \frac{x_j - x_{j-1}}{x_j - x_1} = \sum_{j=2}^{n'} \frac{x_j - x_{j-1}}{x_j - x_1} + \sum_{j=n'+1}^{n} \frac{x_j - x_{j-1}}{x_j - x_1} .$$

As $n \to \infty$, we also have $n' \to \infty$. Therefore, following the analysis in the proof of Theorem 4.1, we have

$$\mathbb{E}\left[\sum_{j=2}^{n'} \frac{x_j - x_{j-1}}{x_j - x_1}\right] = \ln[np(a)(b-a)] + 1 + \gamma + o(1) .$$

By the definition of Riemann integral, we have

$$\mathbb{E}\left[\sum_{j=n'+1}^{n} \frac{x_j - x_{j-1}}{x_j - x_1}\right] = \mathbb{E}\left[\int_{x_{n'}}^{x_n} \frac{1}{x_1 - t} \, dt\right] + o(1)$$

$$= \mathbb{E}[\ln(x_n - x_1) - \ln(x_{n'} - x_1)] + o(1) .$$

Following the analysis in the proof of Theorem 4.1, we have

$$\mathbb{E}[\ln(x_n - x_1) - \ln(x_{n'} - x_1)] + o(1) = \ln(d-a) - \ln(c-a) + o(1) .$$

2. For $x \in (a, b)$, let $i' = \max\{i : x_i \le x\}$ and $n' = \max\{i : x_i \le b\}$. Similarly, we have

$$\bar{h}(x; D) = \bar{h}(x_{i'}) + o(1)$$

and

$$\bar{h}(x_{i'}) = \sum_{j=2}^{i'} \frac{x_j - x_{j-1}}{x_{i'} - x_{j-1}} + \sum_{j=i'+1}^{n'} \frac{x_j - x_{j-1}}{x_j - x_{i'}} + \sum_{j=n'+1}^{n} \frac{x_j - x_{j-1}}{x_j - x_{i'}} .$$

Following the analysis in the analysis of $x \in (-\infty, a)$, we have

$$\mathbb{E}\left[\sum_{j=2}^{i'} \frac{x_j - x_{j-1}}{x_{i'} - x_{j-1}}\right] = \ln[np(x_{i'})(x_{i'} - a)] + 1 + \gamma + o(1) ,$$

$$\mathbb{E}\left[\sum_{j=i'+1}^{n'} \frac{x_j - x_{j-1}}{x_j - x_{i'}}\right] = \ln[np(x_{i'})(x_{n'} - x_{i'})] + 1 + \gamma + o(1) ,$$

$$\mathbb{E}\left[\sum_{j=n'+1}^{n} \frac{x_j - x_{j-1}}{x_j - x_{i'}}\right] = \ln(d - x_{i'}) - \ln(c - x_{i'}) + o(1) .$$

The proof of this case is complete.

3. For $x = b$, the analysis is the same as the analysis for $x = a$.

4. For $x \in (b, c)$, by Theorem A.2, we have

$$\bar{h}(x; D) = \bar{h}(b; D) + \frac{x - b}{c - b} \left(\bar{h}(c; D) - \bar{h}(b; D)\right) .$$

5. For $x = c$, the analysis is the same as the analysis for $x = a$.

6. For $x \in (c, d)$, the analysis is the same as the analysis for $x \in (a, b)$.

7. For $x \in [d, \infty)$, the analysis is the same as the analysis for $x \in (-\infty, a]$. $\square$

## C. Further Elaboration of Assumption 5.2

Recall that we assumed $\kappa > \Omega(\sqrt{n+3})$ in Assumption 5.2. Here, we will show that $\kappa > \Omega(\sqrt{n})$ is commonly satisfied in practice. Note that any smooth distribution can be decomposed into a mixture of uniform distributions. If we can extract a uniform component from a continuous distribution, we can establish a lower bound for $\kappa$. Note that points outside this component tend to increase $\kappa$. We will focus on the uniform distribution, for which we have the following proposition.

**Proposition C.1.** *Let $X_1, \ldots, X_n, n > 3$ be i.i.d. random variables from $\mathcal{U}[0, 1]$. Then, with probability $1 - O(1/n^{1/2})$, the following holds:*

$$\kappa \geq \frac{1}{2}\sqrt{n} \ .$$

**Proof** Let $L = \min_i |X_{i+1} - X_i|$. We first show that

$$\mathbb{E}[L] = 1/(n^2 - 1) \ .$$

By the symmetry of order statistics, we have

$$\Pr[L > t] = n! \Pr[L > t, X_1 < \cdots < X_n] \ .$$

Observing that $\Pr[L > t, X_1 < \cdots < X_n]$ equals the volume of the set

$$S = \{(x_1, \ldots, x_n) \in [0, 1]^n \mid x_i + t \leq x_{i+1}, i = 1, \ldots, n - 1\} \ .$$

Applying the transformation

$$(y_1, \ldots, y_n) = (x_1, x_2 - t, x_3 - 2t, \ldots, x_n - (n - 1)t) \ ,$$

which is volume-preserving, gives a new set

$$S' = \{(x_1, \ldots, x_n) \in [0, 1 - (n - 1)t]^n \mid x_i + t \leq x_{i+1}, i = 1, \ldots, n - 1\} \ .$$

Again by the symmetry of order statistics, we have

$$\Pr[L > l, X_1 < \cdots < X_n] = n! \, \mathrm{Vol}(S')$$
$$= n! \, \frac{1}{n!}(1 - (n - 1)t)^n = (1 - (n - 1)t)^n \ .$$

Therefore, we have

$$\mathbb{E}[L] = \int_0^{1/(n-1)} \Pr[L > t] \, dt = \int_0^{1/(n-1)} (1 - (n - 1)t)^n \, dt = \frac{1}{n^2 - 1} \ .$$

By Markov's inequality, we have

$$\Pr\left[L > \frac{1}{n^2 - 1} + \frac{1}{n^{3/2}}\right] \leq \frac{n^{3/2}}{n^2 - 1} = O(1/n^{1/2}) \ .$$

Observing that

$$\Pr[\max_i X_i - \min_i X_i \leq 1/2] \leq \frac{c}{2^n} \ ,$$

for some $c > 0$, which implies that

$$\Pr\left[U \leq \frac{1}{2n}\right] \leq \frac{c}{2^n} \ ,$$

Therefore, we have

$$\Pr\left[\frac{U}{L} \leq \frac{\frac{1}{2n}}{\frac{1}{(n^2-1)} + \frac{1}{n^{3/2}}}\right] \leq O(n^{-\frac{1}{2}} + 2^{-n}) = O(n^{-\frac{1}{2}}) \ ,$$

which completes the proof. □

## D. Proofs of Theorems 5.3-5.5

We prove the theorems about marginal single anomalies here, including Theorems 5.3-5.5.

### D.1. Proof of Theorem 5.3

To avoid ambiguity, we denote by $U_{\mathrm{ms}} = U$ and $L_{\mathrm{ms}} = L$ following the definitions in Definition 5.1. Suppose that $x_2 - x_1 > U_{\mathrm{ms}} \cdot \kappa$. We prove the result by showing that for all $j > 1$, the following holds:

$$\bar{h}(x_1; x_{1:n}) < \sup_{x_{1:n}} \bar{h}(x_1; x_{1:n}) \leq \inf_{j>1, x_{1:n}} \bar{h}(x_j; x_{1:n}) \leq \bar{h}(x_j; x_{1:n}) ,$$

where we take $\sup$ and $\inf$ instead of $\max$ and $\min$ because the maximum and minimum may not exist when

$$x_2 - x_1 > U_{\mathrm{ms}} \cdot \kappa .$$

Note that

$$\bar{h}(x_1; x_{1:n}) < \sup_{x_{1:n}} \bar{h}(x_1; x_{1:n}) \quad \text{and} \quad \inf_{j>1, x_{1:n}} \bar{h}(x_j; x_{1:n}) \leq \bar{h}(x_j; x_{1:n})$$

are trivial. Then it suffices to show that

$$\sup_{x_{1:n}} \bar{h}(x_1; x_{1:n}) \leq \inf_{j>1, x_{1:n}} \bar{h}(x_j; x_{1:n}) .$$

For convenience, we define

$$s_i \triangleq x_{i+1} - x_i, i \leq n - 1 .$$

Apparently, assigning $(x_1, \ldots, x_n)$ is equivalent to assigning $(s_1, \ldots, s_{n-1})$. From the condition $U_{\mathrm{ms}} = \max_{i \geq 2} |x_{i+1} - x_i|$ and $x_2 - x_1 > U_{\mathrm{ms}} \cdot \kappa$, we have

$$s_1 > U_{\mathrm{ms}} \cdot \kappa \geq U_{\mathrm{ms}} \geq s_j, \ \forall j > 1 .$$

We assert that $\inf_{j>1, x_{1:n}} \bar{h}(x_j; x_{1:n})$ is achieved at $j = n$, i.e.,

$$\inf_{j>1, x_{1:n}} \bar{h}(x_j; x_{1:n}) = \inf_{x_{1:n}} \bar{h}(x_n; x_{1:n}) .$$

Otherwise, there exists a $j_0$ satisfying $1 < j_0 < n$ such that

$$\inf_{x_{1:n}} \bar{h}(x_{j_0}; x_{1:n}) = \inf_{j>1, x_{1:n}} \bar{h}(x_j; x_{1:n}) < \inf_{x_{1:n}} \bar{h}(x_n; x_{1:n}) .$$

By Lemma 3.4, we have

$$\bar{h}(x_{j_0}; x_{1:n}) = \frac{s_1}{s_1 + \cdots + s_{j_0-1}} + \frac{s_2}{s_2 + \cdots + s_{j_0-1}} + \cdots + \frac{s_{j_0-1}}{s_{j_0-1}}$$
$$+ \frac{s_{j_0}}{s_{j_0}} + \frac{s_{j_0+1}}{s_{j_0} + s_{j_0+1}} + \cdots + \frac{s_{n-1}}{s_{j_0} + \cdots + s_{n-1}} .$$

Notice that

$$\frac{s_{j_0}}{s_{j_0}} + \frac{s_{j_0+1}}{s_{j_0} + s_{j_0+1}} + \cdots + \frac{s_{n-1}}{s_{j_0} + \cdots + s_{n-1}}$$
$$> \frac{s_{j_0}}{s_{j_0} + s_1 + \cdots + s_{j_0-1}} + \frac{s_{j_0+1}}{s_{j_0+1} + s_{j_0} + s_1 + \cdots + s_{j_0-1}}$$
$$+ \cdots + \frac{s_{n-1}}{s_{n-1} + \cdots + s_{j_0} + s_1 + \cdots + s_{j_0-1}}$$
$$= \sum_{i=j_0}^{n-1} \frac{s_i}{s_i + \cdots + s_{j_0} + s_1 + \cdots + s_{j_0-1}} .$$

Therefore, we have

$$\bar{h}(x_{j_0}; x_{1:n}) > \sum_{i=1}^{j_0-1} \frac{s_i}{s_i + \cdots + s_{j_0-1}} + \sum_{i=j_0}^{n-1} \frac{s_i}{s_i + \cdots + s_{j_0} + s_1 + \cdots + s_{j_0-1}} .$$

If we reassign $(s'_1, \ldots, s'_{n-1})$ with $(s_{n-1}, \ldots, s_{j_0}, s_1, \ldots, s_{j_0-1})$ and reassign the corresponding $(x'_1, \ldots, x'_n)$ with $s'_i$, then we have

$$\bar{h}(x'_n; x'_{1:n}) = \sum_{i=1}^{j_0-1} \frac{s_i}{s_i + \cdots + s_{j_0-1}} + \sum_{i=j_0}^{n-1} \frac{s_i}{s_i + \cdots + s_{j_0} + s_1 + \cdots + s_{j_0-1}} < \bar{h}(x_{j_0}; x_{1:n}) \, ,$$

which conflicts with the condition that $\bar{h}(x_{j_0}; x_{1:n})$ is the minimal. To this end, it suffices to show that

$$\sup_{x_{1:n}} \bar{h}(x_1; x_{1:n}) < \inf_{x_{1:n}} \bar{h}(x_n; x_{1:n}) \, .$$

Recall that

$$\bar{h}(x_1; x_{1:n}) = \frac{s_1}{s_1} + \frac{s_2}{s_1 + s_2} + \cdots + \frac{s_{n-1}}{s_1 + \cdots + s_{n-1}} \triangleq \tilde{h}(x_1; s_1, \ldots, s_{n-1}) \, ,$$

which is decreasing with respect to $s_1$ and increasing with respect to $s_{n-1}$, implying that

$$\sup_{x_{1:n}} \bar{h}(x_1; x_{1:n})$$

$$= \sup_{s_2, \ldots, s_{n-2}} \tilde{h}(x_1; U_{\mathrm{ms}} \cdot \kappa, s_2, \ldots, s_{n-2}, U_{\mathrm{ms}})$$

$$= \sup_{s_2, \ldots, s_{n-2}} \frac{U_{\mathrm{ms}} \cdot \kappa}{U_{\mathrm{ms}} \cdot \kappa} + \frac{s_2}{U_{\mathrm{ms}} \cdot \kappa + s_2} + \cdots + \frac{U_{\mathrm{ms}}}{U_{\mathrm{ms}} \cdot \kappa + s_2 + \cdots + s_{n-2} + U_{\mathrm{ms}}} \, .$$

We now consider the following optimization problem

$$\max_{s_2, \ldots, s_{n-2}} \frac{U_{\mathrm{ms}} \cdot \kappa}{U_{\mathrm{ms}} \cdot \kappa} + \frac{s_2}{U_{\mathrm{ms}} \cdot \kappa + s_2} + \cdots + \frac{U_{\mathrm{ms}}}{U_{\mathrm{ms}} \cdot \kappa + s_2 + \cdots + s_{n-2} + U_{\mathrm{ms}}} \, ,$$
$$\text{s.t.} \quad L_{\mathrm{ms}} \le s_i \le U_{\mathrm{ms}}, \quad \forall i \, ,$$

which has a Lagrangian function as follows:

$$L(s_2, \ldots, s_{n-2}) = \frac{U_{\mathrm{ms}} \cdot \kappa}{U_{\mathrm{ms}} \cdot \kappa} + \frac{s_2}{U_{\mathrm{ms}} \cdot \kappa + s_2} + \cdots + \frac{U_{\mathrm{ms}}}{U_{\mathrm{ms}} \cdot \kappa + s_2 + \cdots + s_{n-2} + U_{\mathrm{ms}}}$$
$$+ \mu_2 (L_{\mathrm{ms}} - s_2) + \cdots + \mu_{n-2} (L_{\mathrm{ms}} - s_{n-2})$$
$$+ \mu'_2 (s_2 - U_{\mathrm{ms}}) + \cdots + \mu'_{n-2} (s_{n-2} - U_{\mathrm{ms}}) \, .$$

The KKT conditions are

$$\frac{\partial L}{\partial s_i} = \frac{U_{\mathrm{ms}} \cdot \kappa + s_2 + \cdots + s_{i-1}}{(U_{\mathrm{ms}} \cdot \kappa + s_2 + \cdots + s_{i-1} + s_i)^2} - \sum_{i'=i+1}^{n-2} \frac{s_{i'}}{(U_{\mathrm{ms}} \cdot \kappa + s_2 + \cdots + s_{i'})^2}$$

$$- \frac{U_{\mathrm{ms}}}{U_{\mathrm{ms}} \cdot \kappa + s_2 + \cdots + s_{n-2} + U_{\mathrm{ms}}} - \mu_i + \mu'_i$$

$$= 0 \, ,$$

$$(L_{\mathrm{ms}} - s_i) \le 0 \, , \quad (s_i - U_{\mathrm{ms}}) \le 0 \, , \quad \mu_i \le 0 \, , \quad \mu'_i \le 0$$
$$\mu_i (L_{\mathrm{ms}} - s_i) = 0 \, , \quad \mu'_i (s_i - U_{\mathrm{ms}}) = 0 \, .$$

We have the following KKT point:

$$(s_2, \ldots, s_{n-2}) = (U_{\mathrm{ms}}, \ldots, U_{\mathrm{ms}}) \, .$$

To verify this KKT point, we first observe that

$$\sum_{i'=i+1}^{n-2} \frac{1}{(U_{\text{ms}} \cdot \kappa + (i'-1)U_{\text{ms}})^2} + \frac{U_{\text{ms}}}{U_{\text{ms}} \cdot \kappa + s_2 + \cdots + s_{n-2} + U_{\text{ms}}}$$

$$= \sum_{i'=i+1}^{n-1} \frac{1}{(U_{\text{ms}} \cdot \kappa + (i'-1)U_{\text{ms}})^2}$$

$$< \sum_{i'=i+1}^{n-1} \left( \frac{1}{U_{\text{ms}} \cdot \kappa + (i'-2)U_{\text{ms}}} - \frac{1}{U_{\text{ms}} \cdot \kappa + (i'-1)U_{\text{ms}}} \right)$$

$$= \frac{1}{U_{\text{ms}} \cdot \kappa + (i-1)U_{\text{ms}}} - \frac{1}{U_{\text{ms}} \cdot \kappa + (n-2) \cdot U_{\text{ms}}} \,,$$

Note that the following holds under Assumption 5.2:

$$\frac{U_{\text{ms}} \cdot \kappa + (i-2)U_{\text{ms}}}{(U_{\text{ms}} \cdot \kappa + (i-1)U_{\text{ms}})^2} - \frac{U_{\text{ms}}}{(U_{\text{ms}} \cdot \kappa + (i-1)U_{\text{ms}})} + \frac{U_{\text{ms}}}{(U_{\text{ms}} \cdot \kappa + (n-2)U_{\text{ms}})} > 0 \,,$$

implying that

$$\mu_i = 0 \quad \text{and} \quad \mu_i' < 0 \,,$$

which satisfies the KKT conditions. Therefore, we have

$$\sup_{x_{1:n}} \bar{h}(x_1; x_{1:n}) = \sup_{s_2, \ldots, s_{n-1}} \tilde{h}(x_1; s_1, s_2, \ldots, s_{n-1}) = \tilde{h}(x_1; U_{\text{ms}} \cdot \kappa, \underbrace{U_{\text{ms}}, \ldots, U_{\text{ms}}}_{n-2}) \,.$$

Similarly, we have

$$\inf_{x_{1:n}} \bar{h}(x_n; x_{1:n}) = \inf_{s_1, \ldots, s_{n-1}} \tilde{h}(x_n; s_1, s_2, \ldots, s_{n-1}) = \tilde{h}(x_n; \underbrace{L_{\text{ms}}, \ldots, L_{\text{ms}}}_{n-2}, U_{\text{ms}}) \,.$$

By Lemma 3.4, we have

$$\sup_{x_{1:n}} \bar{h}(x_1; x_{1:n}) = \tilde{h}(x_1; U_{\text{ms}} \cdot \kappa, \underbrace{U_{\text{ms}}, \ldots, U_{\text{ms}}}_{n-2})$$

$$= \frac{U_{\text{ms}} \cdot \kappa}{U_{\text{ms}} \cdot \kappa} + \frac{U_{\text{ms}}}{U_{\text{ms}} \cdot \kappa + U_{\text{ms}}} + \cdots + \frac{U_{\text{ms}}}{U_{\text{ms}} \cdot \kappa + U_{\text{ms}} + \cdots + U_{\text{ms}} + U_{\text{ms}}}$$

$$\leq \frac{U_{\text{ms}}}{U_{\text{ms}}} + \frac{L_{\text{ms}}}{L_{\text{ms}} + U_{\text{ms}}} + \cdots + \frac{L_{\text{ms}}}{L_{\text{ms}} + \cdots + L_{\text{ms}} + U_{\text{ms}}}$$

$$= \tilde{h}(x_n; \underbrace{L_{\text{ms}}, \ldots, L_{\text{ms}}}_{n-2}, U_{\text{ms}})$$

$$\leq \inf_{x_{1:n}} \bar{h}(x_n; x_{1:n}) \,,$$

which completes the proof. $\qquad\square$

## D.2. Proof of Theorem 5.4

We prove the result by giving an assignment of $(x_1, \ldots, x_n)$ such that when $U_{\text{ms}} \cdot \kappa > x_2 - x_1 > U_{\text{ms}}$,

$$\exists j > 1, \quad \bar{h}(x_1; x_{1:n}) \leq \bar{h}(x_j; x_{1:n}) \,.$$

We assign $x_i$ from the corresponding $s_i$ as follows

$$(s_1, \ldots, s_{n-1}) = (U_{\text{ms}} + \epsilon, U_{\text{ms}}/2, U_{\text{ms}}, \ldots, U_{\text{ms}}) \,,$$

where $\epsilon > 0$ is relatively small compared with $U_{\text{ms}}$. By mathematical induction on $n$, we can show that

$$\tilde{h}(x_1; U_{\text{ms}}, U_{\text{ms}}/2, U_{\text{ms}}, \ldots, U_{\text{ms}}) = \tilde{h}(x_1; 1, 1/2, 1, \ldots, 1)$$
$$> \tilde{h}(x_n; 1, 1/2, 1, \ldots, 1)$$
$$= \tilde{h}(x_n; U_{\text{ms}}, U_{\text{ms}}/2, U_{\text{ms}}, \ldots, U_{\text{ms}})$$

By the continuity of $\tilde{h}$ and $\bar{h}$, there exists an $\epsilon > 0$ such that

$$\tilde{h}(x_1; U_{\text{ms}} + \epsilon, U_{\text{ms}}/2, U_{\text{ms}}, \ldots, U_{\text{ms}}) > \tilde{h}(x_n; U_{\text{ms}} + \epsilon, U_{\text{ms}}/2, U_{\text{ms}}, \ldots, U_{\text{ms}})$$

The proof is complete. $\qquad\square$

### D.3. Proof of Theorem 5.5

Recall that the output of $k$-nearest neighbor is

$$h_{knn}(x; D) \triangleq \frac{1}{k} \sum_{x' \in \mathcal{N}_k(x)} \|x - x'\|_1 \, .$$

Similar to the proof of Theorem 5.3, we prove the result by showing that for all $j > 1$, the following holds:

$$h_{knn}(x_1; x_{1:n}) > \inf_{x_{1:n}} h_{knn}(x_1; x_{1:n}) \geq \sup_{j', x_{1:n}} h_{knn}(x_{j'}; x_{1:n}) \geq h_{knn}(x_j; x_{1:n}),$$

for which the direction of the inequalities is reversed compared to Theorem 5.3. This arises due to the negative correlation between the path length function and the distance to the nearest neighbor. Similarly, we define

$$\tilde{h}_{knn}(x_1; s_1, \ldots, s_{n-1}) \triangleq h_{knn}(x_1; x_{1:n}) \, ,$$

where $s_i \triangleq x_{i+1} - x_1$ denotes the difference between two neighboring points. Then, it is not difficult to show that

$$\inf_{x_{1:n}} h_{knn}(x_1; x_{1:n}) = \inf_{s_{1:n-1}} \tilde{h}_{knn}(x_1; s_{1:n-1})$$
$$= \tilde{h}_{knn}(s_1^\circ, L_{\text{ms}}, \ldots, L_{\text{ms}})$$
$$= \frac{1}{k} \left[ s_1^\circ + s_1^\circ + L_{\text{ms}} + \cdots + s_1^\circ + (k-1)L_{\text{ms}} \right]$$
$$= \frac{1}{k} \left[ \frac{k(k+1)s_1^\circ}{2} + \frac{k(k-1)}{2} L_{\text{ms}} \right]$$
$$= \frac{k+1}{2} s_1^\circ + \frac{k-1}{2} L_{\text{ms}}$$
$$= \frac{k+1}{2} U_{\text{ms}} \, ,$$

where $s_1^\circ = U_{\text{ms}} + (k-1)(U_{\text{ms}} - L_{\text{ms}})/2$. Similarly, we have

$$\sup_{x_{1:n}} h_{knn}(x_n; x_{1:n}) = \sup_{s_{1:n-1}} \tilde{h}_{knn}(x_n; s_{1:n-1})$$
$$= \tilde{h}_{knn}(x_n, U_{\text{ms}}, \ldots, U_{\text{ms}})$$
$$= \frac{1}{k} (U_{\text{ms}} + 2U_{\text{ms}} + \cdots + kU_{\text{ms}})$$
$$= \frac{k+1}{2} U_{\text{ms}}$$
$$= \inf_{x_{1:n}} h_{knn}(x_1; x_{1:n}) \, .$$

$\qquad\square$

# E. Proofs of Theorems 5.6-5.7

We will prove Theorems 5.6-5.7 in this section.

## E.1. Proof of Theorem 5.6

Before that, we first define $U_{\mathrm{cs}}$ and $L_{\mathrm{cs}}$ as follows:

$$U_{\mathrm{cs}} = \max_{j \neq n_0/2} |x_{j+1} - x_j| \quad L_{\mathrm{cs}} = \min_{j \neq n_0/2} |x_{j+1} - x_j| \ .$$

Similar to the proof of Theorem 5.3, we define

$$\tilde{h}(x_i; s_1, \ldots, s_{n-1}) = \bar{h}(x_i; x_{1:n}) \ ,$$

where $x_i \in D$ is an arbitrary element in $D$. For convenience, we define

$$\theta = \min\{x_{m+1} - x_m, x_{m+2} - x_{m+1}\} \ .$$

We will prove the sufficiency and necessity of $\theta = \Theta(\sqrt{n_0})$ separately.

**Sufficiency:** There exists a constant $c_1$ such that when $\theta > c_1 \cdot \sqrt{n_0}$, we have

$$\forall j, \ \bar{h}(x_{n_0/2+1}; x_{1:n}) < \min_j \bar{h}(x_j; x_{1:n}) \ .$$

By the definition of $\tilde{h}$

$$
\begin{aligned}
\bar{h}(x_{n_0/2+1}; x_{1:n}) &= \tilde{h}(x_{n_0/2+1}; s_1, \ldots, s_{n-1}) \\
&< 2 \sup_{s_{1:n_0/2}} \tilde{h}(s_{n_0/2}; s_1, \ldots, s_{n_0/2}) \\
&\leq 2\tilde{h}(s_{n_0/2}; U_{\mathrm{cs}}, \ldots, U_{\mathrm{cs}}, \theta) \\
&= 2 \left( \frac{\theta}{\theta} + \frac{U_{\mathrm{cs}}}{\theta + U_{\mathrm{cs}}} + \frac{U_{\mathrm{cs}}}{\theta + 2U_{\mathrm{cs}}} + \cdots + \frac{U_{\mathrm{cs}}}{\theta + (n_0/2 - 1)U_{\mathrm{cs}}} \right) \\
&\leq 2 + 2\ln \left[ 1 + \frac{n_0/2 - 1}{\theta/U_{\mathrm{cs}}} \right] \ .
\end{aligned}
$$

Similarly, we have

$$
\begin{aligned}
\min_j \bar{h}(x_j; x_{1:n}) &\geq \inf_{j, s_{1:n}} \bar{h}(x_j; x_{1:n}) \\
&= \inf_{s_{1:n-1}} \tilde{h}(x_1; s_{1:n-1}) \text{ (follows the analysis in Appendix D.1)} \\
&\geq \inf_{s_{1:n_0/2-1}} \tilde{h}(x_1; s_{1:n_0/2-1}) \\
&= \frac{U_{\mathrm{cs}}}{U_{\mathrm{cs}}} + \frac{L_{\mathrm{cs}}}{U_{\mathrm{cs}} + L_{\mathrm{cs}}} + \frac{L_{\mathrm{cs}}}{U_{\mathrm{cs}} + 2L_{\mathrm{cs}}} + \cdots + \frac{L_{\mathrm{cs}}}{U_{\mathrm{cs}} + (n_0/2 - 1)L_{\mathrm{cs}}} \\
&\geq 1 + \ln \left[ 1 + \frac{n_0/2 - 1}{U_{\mathrm{cs}}/L_{\mathrm{cs}} + 1} \right] \ .
\end{aligned}
$$

Thus, we expect that

$$
\begin{aligned}
2 + 2\ln \left[ 1 + \frac{n_0/2 - 1}{\theta/U_{\mathrm{cs}}} \right] &= \ln \left[ e^2 \left( 1 + \frac{n_0/2 - 1}{\theta/U_{\mathrm{cs}}} \right)^2 \right] \\
&\leq \ln \left[ e \left( 1 + \frac{n_0/2 - 1}{U_{\mathrm{cs}}/L_{\mathrm{cs}} + 1} \right) \right] = 1 + \ln \left[ 1 + \frac{n_0/2 - 1}{U_{\mathrm{cs}}/L_{\mathrm{cs}} + 1} \right] \ ,
\end{aligned}
$$

which holds if $\theta \geq \Omega(\sqrt{n_0})$.

**Necessity:** If $\theta < o(\sqrt{n_0})$, there exists an assignment of $x_1, \ldots, x_n$ such that

$$\exists j, \ \bar{h}(x_{n_0/2+1}; x_{1:n}) \geq \bar{h}(x_j; x_{1:n}) .$$

We prove this by giving an example of such an assignment as follows:

$$x_1 = 1, \quad x_2 = 2, \quad \ldots, \quad x_{n_0/2} = n_0/2, \quad x_{n_0/2+1} = n_0/2 + \theta,$$

$$x_{n_0/2+2} = n_0/2 + 2\theta, \quad x_{n_0/2+3} = n_0/2 + 2\theta + 1, \quad \ldots, \quad x_n = n_0/2 + 2\theta + n_0/2 ,$$

or equivalently,

$$(s_1, \ldots, s_{n-1}) = (1, \ldots, 1, \theta, \theta, 1, \ldots, 1) .$$

Then we have

$$\tilde{h}(x_{n_0/2+1}) = 2 \left( \frac{\theta}{\theta} + \frac{1}{\theta+1} + \frac{1}{\theta+2} + \cdots + \frac{1}{\theta + (n_0/2 - 1)} \right)$$

$$\geq 2 + 2\ln \left( 1 + \frac{n_0/2 - 1}{\theta + 1} \right)$$

$$\geq \ln \left( \Omega(n_0^2/\theta^2) \right)$$

and

$$\tilde{h}(x_1) = 1 + \frac{1}{2} + \frac{1}{3} + \cdots + \frac{1}{n_0/2 - 1} + \frac{\theta}{n_0/2 - 1 + \theta} +$$

$$\frac{\theta}{2\theta + n_0/2 - 1} + \frac{1}{n_0/2 - 1 + 2\theta + 1} + \cdots + \frac{1}{n_0/2 - 1 + 2\theta + n_0/2 - 1}$$

$$\leq 1 + \frac{1}{2} + \frac{1}{3} + \cdots + \frac{1}{n_0/2 - 1} + 2 + \ln(2)$$

$$\leq 3 + \ln(2) + \ln \left( n_0/2 - 1 \right)$$

$$= \ln(O(n_0))$$

Therefore, if $\theta = o(\sqrt{n_0})$, we have

$$\tilde{h}(x_{n_0/2+1}) = \omega(\tilde{h}(x_1)) ,$$

implying that when $n_0$ is large enough

$$\bar{h}(x_{n_0/2+1}; x_{1:n}) > \bar{h}(x_1; x_{1:n}) .$$

$\square$

### E.2. Proof of Theorem 5.7

For convenience, we let $k$ be an even number; otherwise, choosing $k + 1$ instead of $k$ works similarly. Following the proof of Theorem 5.5, we have

$$\min_{s_{1:n-1}} \tilde{h}_{knn}(x_{n_0/2+1}; s_1, \ldots, s_{n-1}) = \tilde{h}_{knn}(x_{n_0/2+1}; \underbrace{L_{cs}, \ldots, L_{cs}}_{n_0/2-1}, \theta, \theta, \underbrace{L_{cs}, \ldots, L_{cs}}_{n_0/2-1})$$

$$= \frac{2}{k} \left( \theta + \theta + L_{cs} + \ldots \theta + (k/2 - 1) L_{cs} \right)$$

$$= \frac{2}{k} \left( k\theta/2 + \frac{k/2(k/2 - 1)}{2} L_{cs} \right)$$

$$= \theta + \frac{k/2 - 1}{2} L_{cs} .$$

Similarly, we have

$$\max_{s_{1:n-1}} \tilde{h}_{knn}(x_1; s_1, \ldots, s_{n-1}) = \tilde{h}_{knn}(x_1; \underbrace{U_{cs}, \ldots, U_{cs}}_{n_0/2-1}, \theta, \theta, \underbrace{U_{cs}, \ldots, U_{cs}}_{n_0/2-1},)$$

$$= \frac{1}{k} \left( U_{cs} + 2U_{cs} \cdots + kU_{cs} \right)$$

$$= \frac{k+1}{2} U_{cs} .$$

Then, when $\theta > \frac{k+1}{2} U_{cs} - \frac{k/2-1}{2} L_{cs} = \Omega(k)$, we have $\tilde{h}_{knn}(x_{n_0/2+1}) > \max_j \tilde{h}_{knn}(x_j)$. The necessity can be proved similarly to Appendix E.1, by giving an example of such an assignment as follows:

$$(s_1, \ldots, s_{n-1}) = (\underbrace{1, \ldots, 1}_{n_0/2-1}, \theta, \theta, \underbrace{1, \ldots, 1}_{n_0/2-1}) .$$

$\square$

# F. Proofs of Theorems 5.8-5.9

Here, we will prove Theorems 5.8-5.9. We begin with the following definitions.

$$U_{mg} = \max_{1 \le j \le n_1-1} |x_{j+1} - x_j|, \quad L_{mg} = \min_{n_1+1 \le j \le n_1+n_0-1} |x_{j+1} - x_j|, \quad \text{and } \theta = x_{n_1+1} - x_{n_1} .$$

Since $n_1$ is assumed to be odd, $n_1 = 2q + 1$ for some $q \in \mathbb{N}^+$.

## F.1. Proof of Theorem 5.8

**Sufficiency:** There exists a constant $c_1$ such that when $\theta \ge c_1 \cdot n_1^2$, we have

$$\max_{1 \le j \le n_1} \bar{h}(x_j; x_{1:n}) \le \min_{n_1+1 \le j \le n_1+n_0} \bar{h}(x_j; x_{1:n}) .$$

Following the analysis in Appendix D.1, we have

$$\max_{1 \le j \le n_1} \bar{h}(x_j; x_{1:n}) = \bar{h}(x_{q+1}; x_{1:n}) \quad \text{and} \quad \min_{2q+2 \le j \le 2q+1+n_0} \bar{h}(x_j; x_{1:n}) = \bar{h}(x_n; x_{1:n}) .$$

Then, it suffices to show that

$$\sup_{x_{1:n}} \bar{h}(x_{q+1}; x_{1:n}) \le \inf_{x_{1:n}} \bar{h}(x_n; x_{1:n}) .$$

Similar to the analysis in Appendix D.1, we have

$$\sup_{x_{1:n}} \bar{h}(x_{2q+1}; x_{1:n}) = \sup_{s_{1:q}} \tilde{h}(x_{2q+1}; s_{1:q}) + \sup_{s_{q+1:2q+n_0}} \tilde{h}(x_{2q+1}; s_{q+1:2q+n_0})$$

$$= \tilde{h}(x_{q+1}; \underbrace{U_{mg}, \ldots, U_{mg}}_{q-1}, L_{mg}, L_{mg}, \underbrace{U_{mg}, \ldots, U_{mg}}_{q-1}, \theta, \underbrace{U_{mg}, \ldots, U_{mg}}_{n_0-2})$$

$$= 2 \left( \frac{L_{mg}}{L_{mg}} + \frac{U_{mg}}{L_{mg} + U_{mg}} + \cdots + \frac{U_{mg}}{L_{mg} + (q-1)U_{mg}} \right) +$$

$$\frac{\theta}{2L_{mg} + 2(q-1)U_{mg} + \theta} +$$

$$\frac{U_{mg}}{2L_{mg} + 2(q-1)U_{mg} + \theta + U_{mg}} +$$

$$\cdots +$$

$$\frac{U_{mg}}{2L_{mg} + 2(q-1)U_{mg} + \theta + (n_0 - 2)U_{mg}}$$

$$\le \ln O \left( q^2 n_0/(q+\theta) \right) .$$

and

$$\inf_{x_{1:n}} \bar{h}(x_n; x_{1:n}) = \inf_{s_{1:n-1}} \tilde{h}(x_n; s_{1:n-1})$$

$$= \tilde{h}(x_n; \underbrace{L_{\mathrm{mg}}, \ldots, L_{\mathrm{mg}}}_{2q}, \theta, \underbrace{L_{\mathrm{mg}}, \ldots, L_{\mathrm{mg}}}_{n_0-2}, U_{\mathrm{mg}})$$

$$= \frac{U_{\mathrm{mg}}}{U_{\mathrm{mg}}} + \frac{L_{\mathrm{mg}}}{U_{\mathrm{mg}} + L_{\mathrm{mg}}} + \cdots + \frac{L_{\mathrm{mg}}}{U_{\mathrm{mg}} + (n_0 - 2)U_{\mathrm{mg}}} + *$$

$$\geq \ln\left(\Omega\left(n_0\right)\right) ,$$

where $*$ is some positive quantity. Therefore, $\theta = \Omega(q^2) = \Omega(n_1^2)$ suffices to ensure that

$$\sup_{x_{1:n}} \bar{h}(x_{q+1}; x_{1:n}) \leq \inf_{x_{1:n}} \bar{h}(x_n; x_{1:n}) ,$$

implying that

$$\max_{1 \leq j \leq n_1} \bar{h}(x_j; x_{1:n}) \leq \min_{n_1+1 \leq j \leq n_1+n_0} \bar{h}(x_j; x_{1:n}) .$$

**Necessity:** We will show that if $\theta = o(n_1^2)$, then there exists an assignment of $x_{1:n}$ such that

$$\max_{1 \leq j \leq n_1} \bar{h}(x_j; x_{1:n}) > \min_{n_1+1 \leq j \leq n_1+n_0} \bar{h}(x_j; x_{1:n}) .$$

We similarly give the assignment of $x_{1:n}$ as follows:

$$(s_1, \ldots, s_{n_1-1}) = (\underbrace{1, \ldots, 1}_{2q}, \theta, \underbrace{1, \ldots, 1}_{n_0-1}) .$$

Then, we have

$$\tilde{h}(x_{q+1}; s_{1:n-1}) = 2\left(1 + \frac{1}{2} + \ldots \frac{1}{q}\right) + \frac{\theta}{2q+\theta} + \frac{1}{2q+\theta+1} + \cdots + \frac{1}{2q+\theta+n_0-2}$$

$$\geq \ln\left(\Omega\left(q^2 n_0/(q+\theta)\right)\right)$$

and

$$\tilde{h}(x_n; s_{1:n-1}) = 1 + \frac{1}{2} + \frac{1}{3} + \cdots + \frac{1}{n_0-2} + \frac{\theta}{n_0-2+\theta+1}$$

$$+ \frac{1}{n_0-2+\theta+2} + \cdots + \frac{1}{n_0-2+\theta+2q} \leq \ln O\left(n_0\right) .$$

Thus, if $\theta = o(n_1^2)$, then

$$\max_{1 \leq j \leq n_1} \bar{h}(x_j; x_{1:n}) > \min_{n_1+1 \leq j \leq n_1+n_0} \bar{h}(x_j; x_{1:n}) .$$

The proof is complete. $\square$

### F.2. Proof of Theorem 5.9

**Sufficiency:** There exists a constant $c_1$ such that when $\theta \geq c_1 k$, we have

$$\min_{1 \leq j \leq n_1} h_{knn}(x_j; x_{1:n}) \geq \max_{n_1+1 \leq j \leq n_1+n_0} h_{knn}(x_j; x_{1:n}) .$$

Observing that

$$\min_{1 \leq j \leq n_1} h_{knn}(x_j; x_{1:n}) = \inf_{1 \leq j \leq n_1, s_{1:n-1}} \tilde{h}(x_j; s_{1:n-1})$$

$$= \inf_{1 \leq j \leq n_1, s_{1:n_1-1}} \tilde{h}(x_j; s_{1:n_1-1}, \theta, \underbrace{L_{\mathrm{mg}}, \ldots, L_{\mathrm{mg}}}_{n_0-1})$$

$$\geq \frac{1}{k}\left[(k - n_1 - 2)\theta + (k - n_1 - 2)(k - n_1 - 3)L_{\mathrm{mg}}/2\right]$$

$$= \Omega(\theta + k)$$

and

$$\max_{n_1+1\leq j\leq n_1+n_0} h_{knn}(x_j; x_{1:n}) = \sup_{n_1+1\leq j\leq n_1+n_0, s_{1:n-1}} \tilde{h}(x_j; s_{1:n-1})$$

$$= \sup_{n_1+1\leq j\leq n_1+n_0, s_{1:n_1-1}} \tilde{h}(x_n; s_{1:n_1-1}, \theta, \underbrace{U_{\text{mg}}, \ldots, U_{\text{mg}}}_{n_0-1})$$

$$\leq \frac{1}{k}\left(U_{\text{mg}} + 2U_{\text{mg}} + \cdots + kU_{\text{mg}}\right)$$

$$= O(k) .$$

Therefore, $\theta = \Omega(k)$ suffices to ensure that

$$\min_{1\leq j\leq n_1} h_{knn}(x_j; x_{1:n}) \geq \max_{n_1+1\leq j\leq n_1+n_0} h_{knn}(x_j; x_{1:n}) .$$

**Necessity:** We will show that if $\theta = o(k)$, then there exists an assignment of $x_{1:n}$ such that

$$\min_{1\leq j\leq n_1} h_{knn}(x_j; x_{1:n}) < \max_{n_1+1\leq j\leq n_1+n_0} h_{knn}(x_j; x_{1:n}) .$$

We similarly give the assignment of $x_{1:n}$ corresponding to $s_{1:n-1}$ as follows:

$$(s_1, \ldots, s_{n-1}) = (\underbrace{L_{\text{mg}}, \ldots, L_{\text{mg}}}_{n_1-1}, \theta, \underbrace{L_{\text{mg}}, \ldots, L_{\text{mg}}}_{k-n_1-2}, \underbrace{U_{\text{mg}}, \ldots, U_{\text{mg}}}_{n-k+1}) .$$

Then, we have

$$\min_{1\leq j\leq n_1} h_{knn}(x_j; x_{1:n}) \leq \frac{1}{k}\left(L_{\text{mg}} + 2L_{\text{mg}} + \cdots + (n_1-1)L_{\text{mg}} + \theta + (\theta + L_{\text{mg}}) + \cdots + (\theta + (k-n_1)L_{\text{mg}})\right)$$

$$\leq \left[\frac{n_1(n_1-1)}{2} + \frac{(k-n_1)(k-n_1-1)}{2}\right]\frac{L_{\text{mg}}}{k} + \theta$$

and

$$\max_{n_1+1\leq j\leq n_1+n_0} h_{knn}(x_j; x_{1:n}) \geq \frac{1}{k}(1 + 2 + \cdots + k)U_{\text{mg}} .$$

Thus, we have

$$\min_{1\leq j\leq n_1} h_{knn}(x_j; x_{1:n}) - \max_{n_1+1\leq j\leq n_1+n_0} h_{knn}(x_j; x_{1:n}) \leq -\Omega(k) + \theta .$$

Once $\theta = o(k)$, we have

$$\min_{1\leq j\leq n_1} h_{knn}(x_j; x_{1:n}) - \max_{n_1+1\leq j\leq n_1+n_0} h_{knn}(x_j; x_{1:n}) \leq -\Omega(k) < 0 .$$

The proof is complete. $\square$

