# OpenReview forum: "Theoretical Investigation on Inductive Bias of Isolation Forest"
_ICML.cc/2026/Conference — ICML 2026 regular_

### Official Review · Reviewer_xV8i · 2026-02-25

**Soundness:** 4
**Presentation:** 4
**Significance:** 3
**Originality:** 4
**Overall Recommendation:** 5
**Confidence:** 3

**Summary:**

The authors present a theoretical analysis of the inductive bias of Isolation Forest, with a primary focus on univariate anomaly detection. The authors break down theoretical groundings for expected depth based on random walks. They provide asymptotic, probabilistic guarantees for iForest detection of different types of anomalies for the univariate case, and extend to multivariate cases under the assumption of independence of features. They compare with the inductive biases of k-NN-based anomaly detection.

**Compliance With Llm Reviewing Policy:**

Affirmed.

**Final Justification:**

My main concerns were addressed. Thus, I will keep my acceptance score.

**Key Questions For Authors:**

It is not entirely clear how the empirical studies tie in to the theoretical work. Providing empirical \kappa, \delta values would help. Can you provide additional simulation studies to show asymptotic probabilistic behavior?

Although you mention that the independence assumption is often invoked in similar works, how does a lack of independence affect the key results of your work?

**Limitations:**

yes

**Strengths And Weaknesses:**

The authors provide a strong, well-thought-out background on the importance of this work. The formulation of strong, theoretical guarantees of iForest under certain conditions (univariate, independent multivariate, continuous-valued distributions). Each theorem is followed by an intuitive description of its importance, which helps aid the flow of the paper and untangle the complexities of the nuanced work.

In full transparency, I have not reviewed the proofs of the theorems in the appendices. The theorems (at least the univariate ones), seem intuitive enough. The authors do a great job of providing the mathematical underpinnings for a seemingly simple algorithm.

The presentation is well-structured with only minor changes needed to improve readability.

The problem is of significant importance. Many forest-based methods are taken as black-box models, which further need theoretical justification.

Some minor points of improvement:

Tie the results regarding tree depth back to the original anomaly score formulation presented in Liu et al., (2008).

Define the tree depth (h(x)) before it is mentioned in this original formulation (line 125, left column, page 3).

The notations x_{l_{t}} and x_{r_{t}} are introduced, but then a different formulation (without t) is used in the subsequent Theorem.

Define what is meant by absorbing point, (x_i, x_i)

---

> ### Author Rebuttal · Authors · 2026-03-30
>
> # Rebuttal
>
> We sincerely thank Reviewer xV8i for the positive comments and thoughtful review. We are grateful for the recognition that our paper provides "a strong, well-thought-out background," "strong, theoretical guarantees of iForest," and that "the authors do a great job of providing the mathematical underpinnings for a seemingly simple algorithm." We address the reviewer's suggestions and concerns below.
>
> ---
>
> **Suggestion 1**: About "Tie results back to anomaly score in Liu et al. (2008)"
>
> **Response:** We thank the reviewer for this constructive suggestion. It is straightforward to formally link our results back to the original metric as expected depth is equivalent to the anomaly score. We will include this analysis in the revised manuscript to ensure alignment with the anomaly score formulation in Liu et al. (2008).
>
> ---
>
> **Suggestion 2**: About "Some improvements to the presentation," including defining $h(x)$, clarifying $x_{l_t}$/$x_{r_t}$ notation, and defining absorbing point $(x_i, x_i)$.
>
> **Response**: We sincerely appreciate the reviewer's detailed review. We will carefully double-check the notations and ensure the rigour and readability of our paper.
>
> ---
>
> **Question 1**: About providing empirical $\kappa$, $\delta$ values to tie the empirical studies to the theoretical work, and additional simulation studies to show asymptotic probabilistic behavior.
>
> **Response**: We thank the reviewer for this constructive suggestion. We have added the empirical studies, which are available [here](https://anonymous.4open.science/r/icml2026-submission9370-supplementary-experiment-6F1E/out/main.pdf) **[Figure 3]**. From Figure 3, we can observe that $\kappa$ consistently exceeds the reference line $\sqrt{n + 3}$, which verifies our Assumption 3. Additionally, $\delta$ remains bounded, requiring no further assumptions.
>
> ---
>
> **Question 2**: How does a lack of feature independence affect the key results?
>
> **Response**: We appreciate this insightful question. We agree that extending our analysis to scenarios where the feature independence assumption is violated is an important and meaningful direction.
>
> When features are correlated, the theoretical analysis becomes significantly more complex. We must account for feature dependency and competition, as a split along one dimension alters the candidate split space for all other dimensions. For instance, consider a two-dimensional scenario where normal points form a ring shape and anomalous points are clustered in the center. Here, the features are clearly dependent. If we project these points onto the individual $x^{(1)}$ and $x^{(2)}$ axes, the anomalies appear at the center of the marginal distributions, making them theoretically difficult to detect under our univariate-based analysis in Section 4.2. Empirically, however, iForest successfully isolates these central anomalies. This highlights a discrepancy where simply averaging the expected depths across independent dimensions fails to fully capture the algorithm's joint-feature behavior.
>
> As a preliminary step, we focused on the independent feature setting. This allowed us to naturally extend our one-dimensional results to multivariate cases by averaging the expected depths of the separated features. We acknowledge that this assumption can be restrictive in practice, and analyzing correlated features remains a great challenge. This challenge also occurs in other tree-based methods, such as decision trees [1, 2] and random forests [3] (which assume stronger uniform distribution of features).
>
> To relax the independence assumption in future work, a promising starting point is the bivariate Gaussian distribution, where the degree of feature dependence can be precisely controlled via the correlation coefficient $\rho$. This may be theoretically tractable and can be generalized to higher dimensions. Alternatively, rather than deriving an exact expected depth function, one could analyze the inductive bias by establishing upper and lower bounds on the depth of individual points, and these may benefit from the analysis in one-dimension.
>
> We thank again the reviewer for the valuable comment. We will add a detailed discussion regarding the impact of feature dependence and potential future work in the revised version.
>
> [1] A. T. Kalai and S.-H. Teng. Decision trees are PAC-learnable from most product distributions: A smoothed analysis. ArXiv:0812.0933, 2008.
>
> [2] A. Brutzkus, A. Daniely, E. Malach. ID3 Learns Juntas for Smoothed Product Distributions, ICML, 2020.
>
> [3] E. Scornet, G. Biau, and J. P. Vert. Consistency of random forests. Annals of Statistics, 2015.
>
> ---
>
> Overall, we hope our responses address the reviewer's concerns. We are sincerely grateful for the reviewer's constructive and encouraging feedback, which will undoubtedly improve the quality of our paper.

---

> > ### Author Rebuttal · Reviewer_xV8i · 2026-04-01
> >
> > Thank you for addressing my concerns.

---

### Official Review · Reviewer_3NQt · 2026-03-04

**Soundness:** 3
**Presentation:** 4
**Significance:** 3
**Originality:** 4
**Overall Recommendation:** 5
**Confidence:** 4

**Summary:**

This paper provides a theoretical analysis of Isolation Forest (iForest) by modeling its growth process as a random walk on states defined by node boundaries. The authors derive a closed-form expression for the expected depth function (Theorem 3.5) and use it to compare iForest with k-NN across traditional anomaly scenarios. The main findings are that iForest is less sensitive to central anomalies but more parameter-adaptive than k-NN, and that in the infinite-sample limit, iForest depth captures both local density and global distance to support endpoints (Theorem 5.1).

**Compliance With Llm Reviewing Policy:**

Affirmed.

**Final Justification:**

All three concerns are addressed: the discussion of correlated features (1) and real-world anomaly patterns (2) will strengthen the paper, and the new KDDCup99 experiment (3) concretely illustrates the predicted central-vs-marginal trade-off.

**Key Questions For Authors:**

Can you reply to the weaknesses?

**Limitations:**

Yes

**Strengths And Weaknesses:**

## Main Strengths

- The random walk formulation (Section 3) is elegant and novel. Modeling state transitions via leftmost/rightmost boundaries is a clean assumption that is the key to get the proposed results.

- The paper derives necessary *and* sufficient conditions for anomaly detection across multiple scenarios (Theorems 4.3, 4.6, 4.8), going beyond the "sufficient only" results typical in this literature. This is a strong theoretical contribution.

- Theorem 5.1, characterizing iForest depth as log-density plus log-distance to endpoints, is the paper's most insightful result. It concretely explains how iForest differs from purely density-based or distance-based detectors and could guide practitioners in choosing methods.

- The paper is well-structured and clearly written, with rigorous proofs in the appendix (although I haven't checked them all in details).

## Main Weaknesses

- The main analysis is one-dimensional, and the multi-dimensional extension (Theorem 5.4) requires feature independence (Assumption 5.3). This is a standard and understandable simplification for a first theoretical analysis, but the paper would benefit from a more explicit discussion of this limitation and when/which insights may or may not transfer to correlated features.

- The case studies use stylized anomaly configurations. A brief discussion of how these canonical cases relate to more complex real-world anomaly patterns (e.g., mixed types) would strengthen the paper's from a practical perspective.

- The experiments primarily verify the theoretical predictions on synthetic data. Adding a small set of controlled experiments on real data -- for instance, demonstrating scenarios where iForest predictably outperforms or underperforms k-NN due to the identified inductive biases -- would make the theoretical insights more relevant.

## Suggestions for Improvement

- Consider adding a discussion paragraph on the prospects and challenges of extending the analysis to correlated features, even if a full treatment is left for future work.

- A targeted experiment on a few real datasets (not necessarily a full benchmark suite) illustrating the central-vs-marginal anomaly trade-off predicted by the theory would significantly increase the paper's impact.

---

> ### Author Rebuttal · Authors · 2026-03-30
>
> # Rebuttal
>
> We sincerely thank Reviewer 3NQt for the thorough and positive review. We are especially encouraged by the recognition that:
>
> - "The random walk formulation is **elegant and novel**."
> - Our paper "derives necessary _and_ sufficient conditions for anomaly detection across multiple scenarios, going beyond the 'sufficient only' results typical in this literature. This is a **strong theoretical contribution**."
> - "Theorem 5.1 is the paper's most insightful result. It concretely explains how iForest differs from purely density-based or distance-based detectors and could **guide practitioners** in choosing methods."
> - "The paper is well-structured and clearly written, with **rigorous proofs** in the appendix."
>
> We address the concerns and suggestions below.
>
> ---
>
> **Concern 1**: About the concern "The main analysis is one-dimensional, and the multi-dimensional extension (Theorem 5.4) requires feature independence (Assumption 5.3)" and the suggestion to discuss the prospects and challenges of extending the analysis to correlated features.
>
> **Response**: We appreciate the reviewer's understanding that the independence assumption is a standard simplification for a first theoretical analysis. We agree that extending this framework to correlated features is a critical next step.
>
> As the reviewer rightly points out, introducing feature correlation significantly complicates the analysis. Specifically, splitting at any dimension can alter the candidate split of any other dimension, creating dependency and competition between features. As a preliminary step, assuming feature independence allows us to naturally extend our one-dimensional results to higher dimensions by averaging the expected depths across separated features.
>
> We conjecture that although deriving the exact closed-form expected depth for correlated features is challenging, the property of "probability density + distance to endpoints" still holds for any correlated features, and we are actively investigating this. Moreover, for the inductive bias analysis of iForest, it is alternative to derive the upper and lower bounds of the depth, which we believe can also reduce to the one-dimensional case.
>
> We thank the reviewer for this valuable suggestion. In the revised manuscript, we will explicitly discuss these limitations, challenges, and potential future work.
>
> ---
>
> **Concern 2**: About the concern "The case studies use stylized anomaly configurations" and the suggestion to discuss how these canonical cases relate to more complex real-world anomaly patterns (e.g., mixed types).
>
> **Response**: We agree that discussing how these canonical cases relate to more complex, real-world anomaly patterns will strengthen the practical utility of our paper, and we will add a dedicated discussion elaborating on this extension in the revised version. The extension to more complex patterns naturally follows from our findings. As demonstrated in Theorems 4.3-4.9 and Appendices C-E, the expected depth of any given point is dominated by its neighborhood, whereas the contribution of distant points is largely absorbed into constant terms. In the light of this, our theoretical framework is well-equipped to handle more general and mixed anomaly configurations, as the local geometry around the anomaly remains the primary driver of the isolation depth.
>
> ---
>
> **Concern 3**: About the concern on adding controlled experiments on real data illustrating the central-vs-marginal anomaly trade-off predicted by the theory to make the theoretical insights more relevant.
>
> **Response**: We appreciate this valuable suggestion. To address this concern, we have conducted additional experiments using the HTTP subset of the KDDCup99 dataset. To visualize the comparison between the two algorithms, we applied random projection to reduce the data to 2D. The results can be found [here](https://anonymous.4open.science/r/icml2026-submission9370-supplementary-experiment-6F1E/out/main.pdf) **[Figure 2]**.
>
> As shown, the majority of data points concentrate along a central line (normal instances). iForest successfully detects all anomalies in the upper region, while $k$-NN's performance is highly sensitive to the parameter $k$. Furthermore, iForest is less likely to flag points near the central line as anomalies, as their $x^{(1)}$ coordinates lie in the middle of the distribution. This aligns with our theoretical conclusion: iForest is less sensitive to central anomalies but exhibits greater parameter adaptability compared to $k$-NN. We will include these experiments and discussion in the revised manuscript.
>
> ---
>
> Overall, we are grateful for the constructive suggestions and will incorporate these improvements in the revised version. Thank you again for the thoughtful and encouraging review.

---

> > ### Author Rebuttal · Reviewer_3NQt · 2026-04-02
> >
> > I thank the authors for the thorough response. All three concerns are addressed: the discussion of correlated features (1) and real-world anomaly patterns (2) will strengthen the paper, and the new KDDCup99 experiment (3) concretely illustrates the predicted central-vs-marginal trade-off. I maintain my score.

---

> > > ### Author Response · Authors · 2026-04-04
> > >
> > > Thank you for your acknowledgment. In the revision, we will add the additional experimental results.

---

### Official Review · Reviewer_dNpF · 2026-03-11

**Soundness:** 2
**Presentation:** 3
**Significance:** 3
**Originality:** 3
**Overall Recommendation:** 5
**Confidence:** 4

**Summary:**

This paper contributes new theoretical analyses related to understanding the inductive biases of Isolation Forest (iforest).
Iforest, and ensemble of isolation trees (itree), is a method for unsupervised anomaly detection, where the goal is to identify examples in the training set which are anomalous.
Each itree is generated randomly based on the training set, and the anomaly score for an instance is derived from the depth of the leaf node it belongs to in the tree.
Instances that get isolated early are more likely to be anomalies.
While this algorithm is well-known and a popular choice in practice, theoretical results justifying its efficacy have been limited.
A random walk model for a single itree is proposed, which is used to derive an expression for the expected depth an example in the tree.
Conditions under which iforest successfully identifies different types of anomalies are provided, and compared with kNN-based anomaly detection.
Empirical results show the convergence of the depth function to the theoretical value, and the ability of iforest to identify anomalies in uniformly distributed data.

**Compliance With Llm Reviewing Policy:**

Affirmed.

**Final Justification:**

Major concerns were addressed in rebuttal. Raised my score from weak reject to accept.

**Key Questions For Authors:**

Q1. Does theorem 3.2 describe the growth process of original itree, or a modified version where splits are made at data rather than sampling from continuous uniform distribution?

Q2. (From Figure 5) Does the gap get closer to zero for larger number of trees?

Q3. What does the main takeaway mean: iforest demonstrates greater parameter adaptability compared to kNN?

Q4. How does your expected depth function compare with c(n) from the original paper, and why does there appear to be a discrepancy between them?

**Limitations:**

yes

**Strengths And Weaknesses:**

Strengths:
- The paper provides new insights on the inductive biases of a popular algorithm, iforest.
- Paper is mostly easy to follow.

Weaknesses:

Some of the technical details are not clear, or may contain errors.

- The precise definition of $h$, the central function of interest for the analysis, is not clear.
L120 It is introduced as $h(x;D,\Theta)$, the average depth of $x$ in an itree. $D$ is the dataset, $\Theta$ contains the randomness associated with tree construction. What is meant by average depth is not clear.
Later L210 the arguments of the function change, and now accepts a variable number of arguments.  L566 the $D$ argument is dropped.

- Theorem 3.2 does not seem correct. The end points of the interval containing an example at a leaf node are almost-surely never equal to any of the training examples, since thresholds are generated from the continuous uniform distribution.

- Majority of the results only apply to a single itree with 1-dimensional data. E.g., L198-203: the random walk model does not hold for d>1, as the growth process is much more complicated to write down for d>1.
It is not obvious how the results extend to multiple dimensions, since the formulas either do not depend on d, or are invalid if 1-dimensional data is replaced by vectors.


- One of the two main conclusions (iforest demonstrates greater parameter adaptability compared to kNN) is not clear in its meaning.

- Based on figure 5, it does not look like the observed depths from iforest converge to expected depth suggested by your theory, there may be some bias.
Perhaps this is caused by incorrect random walk model for itree, or simply that more trees are needed to see convergence.

---

> ### Author Rebuttal · Authors · 2026-03-30
>
> # Rebuttal
>
> We sincerely thank Reviewer dNpF for the detailed review and constructive feedback, including the recognition that our paper provides "new insights on the inductive biases of a popular algorithm" and is "mostly easy to follow." We address each concern below.
>
> ---
>
> ### **About the definition of average depth and the notation of $h$** (Ref: W1)
>
> **Response**: Thanks for the thorough review, we are sorry for typos and confusion caused by the notation. We clarify the definitions as follows:
>
> - $h(x; D, \Theta)$ denotes the depth of $x$ in a single iTree. Given $D$ and $\Theta$, $h$ is a deterministic depth function. We will correct the typo in L120 that introduces $h$ as the "average" depth.
> - $\frac{1}{M} \sum_{m=1}^{M} h(x; D, \Theta_m)$: average depth over $M$ trees, the empirical quantity in iForest.
> - $\mathbb{E}_{\Theta}[h(x; D, \Theta)]$: expected depth, the primary focus of our analysis.
>
> We will carefully proofread the manuscript.
>
> ---
>
> ### **About the correctness of Theorem 3.2** (Ref: W2 & Q1)
>
> **Response**: We respectfully clarify that **Theorem 3.2 is correct and describes the exact growth process of the original iTree**, not a modified version. The concern may stem from conflating iForest with Completely Random Forests (CRF) [1], which is similar to iForest but differs in splitting.
>
> While both sample split attributes and values uniformly at random, after splitting at $s \in (x_i, x_{i+1})$, CRF samples the next split from $(x_1, s)$ (**data-independent**), while iForest samples from $(x_1, x_i)$ (**data-dependent**) — interval endpoints are always defined by training samples, not previous split values.
>
> As you noted, the probability of drawing a split value exactly equal to a training sample is zero. However, iForest's splitting rules ensure that interval endpoints are always defined by training samples. Therefore, Theorem 3.2 correctly describes the original iTree.
>
> [1] Leo Breiman. Random Forests. Machine Learning, 45(1):5-32, 2001.
>
> ---
>
> ### **About results limited to one-dimensional data** (Ref: W3)
>
> **Response**: We respectfully argue that the 1D results already constitute a significant contribution — we are the first to derive a closed-form expected depth, establish necessary _and_ sufficient conditions for anomaly detection, and formally characterize iForest's inductive bias. As Reviewer 3NQt recognizes, "this is a standard and understandable simplification for a first theoretical analysis."
>
> We have also considered $d > 1$: under feature independence (Assumption 5.3), our 1D results naturally extend via Theorem 5.4. For the general case with correlated features, the analysis becomes substantially harder due to inter-feature dependency. This is a fundamental challenge shared across the tree-based method literature — e.g., decision tree analyses [1, 2] assume smoothed product distributions, and random forest theory [3] assumes uniform features, both no weaker than our assumption. We will add a dedicated discussion on future directions in the revised paper.
>
> [1] Kalai & Teng. Decision trees are PAC-learnable from most product distributions. ArXiv:0812.0933, 2008.
>
> [2] Brutzkus et al. ID3 Learns Juntas for Smoothed Product Distributions, ICML, 2020.
>
> [3] Scornet et al. Consistency of random forests. Annals of Statistics, 2015.
>
> ---
>
> ### **About "parameter adaptability"** (Ref: W4 & Q3)
>
> **Response**: Our conclusion "parameter adaptability" is based on the decision thresholds in Table 2. iForest's thresholds are _problem-dependent_ (relying purely on data distribution), while kNN's are both _problem-dependent_ and _algorithm-dependent_ (relying heavily on $k$).
>
> In other words, iForest reliably detects anomalies as long as the distance from normal points exceeds a data-driven threshold, without hyperparameter tuning. For kNN, success is highly sensitive to $k$: too large causes over-smoothing, too small leads to false positives. We will expand this in the revision.
>
> ---
>
> ### **About the convergence in Figure 5, Does the gap get closer to zero for larger number of trees?** (Ref: W5 & Q2)
>
> **Response**: Yes, the gap approaches zero as the number of trees increases. The deviation in Figure 5 is due to randomness in tree construction. We have conducted additional experiments with significantly more trees confirming this convergence. Please refer to [here](https://anonymous.4open.science/r/icml2026-submission9370-supplementary-experiment-6F1E/out/main.pdf) [Table 1 and Figure 1] for updated results.
>
> ---
>
> ### **About the comparison with $c(n)$** (Ref: Q4)
>
> **Response**: Our analysis in Sections 3 and 4 is based on a fixed dataset $D$, where the sample size $n$ is given. From the definition of score function $s$, the analysis of $s$ and depth $h$ are equivalent. We will further tie the results back to the original anomaly score formulation, as also suggested by Reviewer xV8i [Suggestion 1].
>
> ---
>
> We hope all concerns have been addressed. Please let us know if you have further questions.

---

> > ### Author Rebuttal · Reviewer_dNpF · 2026-04-02
> >
> > Thank you for your response. My major concern about thm 3.2 stemmed from the explanation about what is being modelled by the random walk, not confusion with CRF. It seemed like you were modelling the intervals that define leaf nodes, which are determined by the split points in original itree. Using your example, the first leaf node in itree captures data between x_1 and s.
> >
> > Examples of what caused the confusion:
> > L148-150: "each tree node forms an interval, and let x_l_t and x_r_t denote the endpoints of the interval containing x_i at time t."
> > L183-186: "Each state corresponds to a point in a two-dimensional coordinate system, representing the leftmost and rightmost boundaries of the tree node containing the target point."
> >
> > After your rebuttal it now seems like you are actually modelling the intervals from which thresholds are sampled. I am happy to raise my score from weak reject to accept conditioned on the promised changes being implemented, and a clearer explanation about thm 3.2 provided.

---

> > > ### Author Response · Authors · 2026-04-04
> > >
> > > Thank you for your acknowledgment. In the revision, we will refine the description of the random walk modeling, especially the interval definitions, to avoid ambiguity and confusion.

---

### Official Review · Reviewer_fCVb · 2026-03-11

**Soundness:** 4
**Presentation:** 4
**Significance:** 3
**Originality:** 3
**Overall Recommendation:** 5
**Confidence:** 4

**Summary:**

The paper does what it says on the tin: it provides a theoretical investigation of the inductive bias of Isolation Forests. In particular, the authors model the growth of isolation trees as a random walk. This allows them to characterize the behavior of the algorithm in a few different situations (e.g., a single anomaly on one of the margins of the distribution, a single anomaly in the center, ...).

They begin with the simplified setting of a single feature and then generalize to multiple attributes by assuming independence between them. The results show that Isolation Forests are very good at isolating outliers when they lie on the margins of the distribution, but have a harder time when they appear between clusters of points.

The authors also compare the behavior of Isolation Forests with k-NN based anomaly detection, highlighting the trade-offs between the two methods. In particular, they show that k-NN is sensitive to the choice of $k$, which controls a trade-off between false positives and false negatives, while the trade-offs in Isolation Forests are mainly tied to the computational cost of running the algorithm. Empirical evidence is provided to support the accuracy of the theoretical analysis.

**Compliance With Llm Reviewing Policy:**

Affirmed.

**Key Questions For Authors:**

It would be very nice to include an analysis similar to the one provided for Assumption 4.2 for the independence assumption between features: to what extent can this assumption be expected to hold in practice? More importantly, how sensitive are the results when the assumption is violated?

**Limitations:**

yes

**Strengths And Weaknesses:**

## Strengths

- the paper is very well written and easy to read;
- the results are novel and interesting, and the analysis provides several useful insights;
- empirical results show strong agreement with the theoretical findings.

## Weaknesses

A few very minor points:

- While the assumption 4.2 is very well motivated, the authors provide little support for the independence assumption.
- I find the usage of $k$ and $\kappa$ sometimes confusing. I understand the difficulty of finding better notation, but it may be preferable to avoid using two very similar symbols for different concepts.
- a minor typo appears on the first page (search for "the the").

## Overall assessment

I would have rated the paper even higher, but this rating requires demonstrating a high impact on at least one AI sub-area. While I appreciate the paper and find the analysis valuable, I cannot bring myself to consider it highly impactful in that sense.

---

> ### Author Rebuttal · Authors · 2026-03-30
>
> # Rebuttal
>
> We sincerely thank Reviewer fCVb for the positive assessment and constructive feedback, including that "the paper is very well written and easy to read," "the results are novel and interesting," and "empirical results show strong agreement with the theoretical findings." We address the concerns below.
>
> ---
>
> **Comment 1**: "While the assumption 4.2 is very well motivated, the authors provide little support for the independence assumption."
>
> **Response**: We appreciate your thoughtful comment. In practice, this assumption can usually be approximately reached by applying decorrelation transformations (e.g., PCA or Mahalanobis-based whitening) as a preprocessing step to align the data closer to this assumption. It is intuitive to take the feature-independence assumption when one extends one-dimensional analyses to multimensional settings.
>
> Analyzing inherently correlated features leads to significant complexity, as the dependency causes "competition" between features, i.e., a split in one dimension significantly changes the candidate splits in other dimensions. This challenge also occurs in other tree-based methods, such as decision trees [1, 2] and random forests [3] (which assume stronger uniform distribution of features). Modeling this complex feature interplay is a promising direction for future work. We will add a detailed discussion of this assumption to the revised manuscript.
>
> [1] A. T. Kalai and S.-H. Teng. Decision trees are PAC-learnable from most product distributions: A smoothed analysis. ArXiv:0812.0933, 2008.
>
> [2] A. Brutzkus, A. Daniely, E. Malach. ID3 Learns Juntas for Smoothed Product Distributions, ICML, 2020.
>
> [3] E. Scornet, G. Biau, and J. P. Vert. Consistency of random forests. Annals of Statistics, 2015.
>
> ---
>
> **Comment 2**: About some confusing notations and typos
>
> **Response**: Thanks for your suggestions. We promise to fix these typos and polish our manuscript carefully in the revised version.
>
> ---
>
> **Comment 3**: About "I would have rated the paper even higher, but this rating requires demonstrating a high impact on at least one AI sub-area. While I appreciate the paper and find the analysis valuable, I cannot bring myself to consider it highly impactful in that sense."
>
> **Response**: Anomaly detection is a fundamental problem in artificial intelligence, with critical applications across diverse domains, including financial fraud detection, network cybersecurity, medical diagnosis, and industrial predictive maintenance. Among the large number of anomaly detection methods, **Isolation Forest (iForest) is one of the most widely used and effective approaches**. Therefore, theoretical studies of iForest are of significant importance. Our main conclusions provide practical guidance for practitioners: if prior knowledge suggests that anomalies are likely to be extreme values (e.g., very large or very small), iForest is an effective and efficient choice; conversely, if anomalies are expected to lie in intermediate regions, $k$-nearest neighbors ($k$-NN) may be more appropriate. **Our insights can help practitioners better apply iForest based on prior knowledge.** Unlike empirical studies that yield case-specific results, our analysis identifies fundamental differences between these methods, enabling more general conclusions.
>
> We also respectfully highlight the following comments from Reviewers 3NQt and xV8i regarding our contributions:
>
> - **Reviewer 3NQt** noted that "Theorem 5.1... concretely explains how iForest differs from purely density-based or distance-based detectors and **could guide practitioners in choosing methods**," and emphasized this is a "**strong theoretical contribution**."
> - **Reviewer xV8i** stated that "the problem is of significant importance" and that "forest-based methods... further need theoretical justification."
>
> We believe our work represents a significant step toward understanding iForest and offers practical utility for model selection. We respectfully request the reviewer to reconsider the impact of our contributions in light of these theoretical advancements.
>
> ---
>
> **Comment 4**: About the sensitivity of the independence assumption
>
> **Response**: When features are correlated, the theoretical analysis becomes significantly more complex. We must account for feature dependency and competition, as a split along one dimension alters the candidate split space for all other dimensions. A promising starting point is the bivariate Gaussian distribution, where the degree of feature dependence can be controlled via the correlation coefficient $\rho$. This may be theoretically tractable and can be generalized to higher dimensions. We will study this in the revision.
>
> ---
>
> Overall, we are deeply grateful for the reviewer's constructive feedback, which will improve the quality of our paper. We hope our responses address the concerns. If you have any further questions, we are happy to discuss them.

---

> > ### Author Rebuttal · Reviewer_fCVb · 2026-04-01
> >
> > > In practice, this assumption can usually be approximately reached by applying decorrelation transformations (e.g., PCA or Mahalanobis-based whitening) as a preprocessing step to align the data closer to this assumption. It is intuitive to take the feature-independence assumption when one extends one-dimensional analyses to multimensional settings.
> >
> > I believe this justification is somewhat too strong, as decorrelation techniques such as whitening remove only second-order dependencies and do not ensure full statistical independence. That said, the underlying motivation is clear: the assumption serves as a simplifying device that makes the theoretical analysis tractable, which might otherwise be infeasible in high-dimensional settings. Given that the empirical results support the proposed approach, I do not consider this issue critical and will not pursue it further.

---

> > > ### Author Response · Authors · 2026-04-04
> > >
> > > Thank you for your acknowledgment and suggestions. In the revision, we will refine the discussion to better position it as a simplifying assumption for tractable analysis and modify current justification.

---

### Decision · Program_Chairs · 2026-04-30

**Decision:**

Accept (regular)

**Comment:**

The reviewers universally agree the paper represents a valuable and interesting contribution to the literature; providing new theoretical insights into a very popular and well recognised model: isolation forest. Good work from the authors, and clearly worthy of publication.